# Semi-brittle rheology and ice dynamics in DynEarthSol3D

Liz C. Logan[1], Luc L. Lavier[2, 3], Eunseo Choi[4], Eh Tan[5], Ginny A. Catania[2, 3]

[1]Institute for Computational and Engineering Science, University of Texas, Austin, 78712, USA
[2]Department of Geological Science, University of Texas, Austin, 78712, USA
[3]Institute for Geophysics, University of Texas, Austin, 78758, USA
[4]Center for Earthquake Research and Information, University of Memphis, Memphis, 38152, USA
[5]Institute of Earth Sciences, Academia Sinica, Taipei, No. 128, Section 2, Taiwan

*Correspondence to*: Liz C. Logan (liz.curry.logan@gmail.com)

**Abstract.** We present a semi-brittle rheology and explore its potential for simulating glacier and ice sheet deformation using a numerical model DynEarthSol3D (DES) in simple, idealized experiments. DES is a finite element solver for the dynamic and quasi-static simulation of continuous media. The experiments within demonstrate the potential for DES to simulate ice failure and deformation in dynamic regions of glaciers, especially at quickly changing boundaries like glacier termini in contact

with the ocean. We explore the effect that different rheological assumptions have on the pattern of flow and failure. We find that the use of a semi-brittle constitutive law is a sufficient material condition to form the characteristic pattern of basal crevasse-aided pinch-and-swell geometry, which is observed globally in floating portions of ice and can often aid in eroding the ice sheet margins in direct contact with oceans.

**Keywords:**

Glacier, ice sheet, numerical modeling, rheology, ice fracture, basal crevasses.

## 1 Introduction

Accurate prediction of global sea-level rise depends critically on numerical models' ability to project the removal of ice from the margins of ice sheets and glaciers under climate change scenarios –

especially those in contact with oceans. In the past five years numerical models have largely risen to the

challenge of simulating the continent-scale, steady state viscous flow of ice, leading to the development of the latest class of ice sheet models that represent ice physics across many flow regimes and in three spatial dimensions. These are often non-linearly viscous, thermo-mechanical models that solve the so-called full-Stokes (FS) equations (e.g., Martin et al., 2004; Gagliardini and Zwinger, 2008; Larour et al., 2012). Models based on shallow ice (SIA) and shallow shelf (SSA) approximations of the FS equations are also in wide use and simulate ice flow well in most areas (e.g. Winkelmann et al., 2011; Lipscomb et al., 2013).

Despite recent advances many pertinent questions in glaciology remain that could potentially be addressed best from a computational perspective, particularly with regard to calving. However representing the smaller scale physics at the heart of this particular problem (i.e., the fracture of crystalline material) often imposes too large a computational cost to remain a tractable problem for many models. Thus, both the FS and SIA/SSA formulations often employ parametrizations for the most physically complicated aspects of their systems. In particular, the failure of ice within many ice sheet models is often treated using Linear Elastic Fracture Mechanics (e.g., Larour et al., 2004), as a rheologically more flexible, time-dependent scalar damage field (e.g., Duddu et al., 2012), or a mixture of the two (Krug et al., 2014, 2015).

Ice rheology has been studied using both geophysical observations and laboratory experiments (Budd & Jacka, 1989; Sammonds et al., 1998; Goldsby and Kohlstedt, 2001; Mahrenholtz and Wu, 1992). Over short time scales ice behaves elastically before yielding or flowing viscously. Over long time scales ice behaves as a viscous fluid for which the viscosity is non-linearly dependent on both temperature and effective stress (Glen, 1955). The resulting constitutive law is called Glen's flow law in glaciological literature and can be written as:

$$\dot{\varepsilon}_e = A\sigma_e^n, \tag{1}$$

where $A$ is a fluidity parameter, $\dot{\varepsilon}_e$ is the effective strain rate (the square root of the second invariant of the full strain rate tensor), $\sigma_e^n$ is the effective stress, and $n$ is typically set to 3. Laboratory experiments also show that ice strain-rate hardens and that it starts to fracture in a brittle manner at high stress or strain rate (Schulson and Duval, 2009). In nature, calving results from the fracture of ice and is a

consequence of ductile and brittle deformation (van der Veen, 1998; Weiss, 2004). Ductile fracture is initiated by the formation of micro-cracks that eventually coalesce to form a macroscopic fracture (seen for example in the Amery Ice Shelf, Bassis et al., 2008), and is a slow process for which weakening by micro-cracks occurs over a prolonged stress plateau. On the other hand, the breaking or damage process for brittle fractures occurs abruptly for a given value of stress and strain-rate (Sammonds et al., 1998; Schulson and Duval, 2009).

Most ice-flow numerical models simulate the long-term (hundreds to thousands of years), large-scale behaviour of ice sheets using a non-linear viscous formulation to calculate the stress tensor (e.g., Larour et al., 2012). Indeed, for simulating long term flow of ice sheets this is an excellent approximation as the Maxwell viscoelastic stress relaxation timescale (time to dissipate elastic stresses) is on the order of a hours to days – depending on local material properties that affect the ice viscosity and shear modulus (MacAyeal & Sergienko, 2013). When simulating ice rupture, however, these models often employ failure criteria developed with elastic underpinnings. For example, Linear Elastic Fracture Mechanics has been a popular and largely accurate criterion for simulating ice fracture when compared to in-situ crevasse measurements (e.g., van der Veen, 1998; Rist et al., 1999; Mottram and Benn, 2009; Luckman et al., 2012; Krug et al., 2014). Particle-based numerical models also show a great deal of promise for the simulation of tidewater glacier calving on an interannual time scale, where ice failure occurs via the breakage of elastic bonds. Bassis and Jacobs (2013) recently modeled the retreat of Helheim glacier using tightly packed particles that interacted elastically and broke once a threshold stress was achieved, often due to the influences of basal topography and buoyancy in floating portions. Astrom et al. (2013) also implemented a particle-based model that included the effects of viscosity by allowing particles joined by elastic beams to form new bonds with nearby particles when stresses were below brittle failure. Thus even though long-term ice flow is approximated well by purely viscous formulations, efforts to simulate ice failure typically incorporate some measure of elastic ice behaviour.

While it is true that the Maxwell viscoelastic relaxation time is on the order of hours to days (very short times scales), the consideration of elastic stresses may prove illuminating and useful in understanding terminus retreat. This retreat depends on a long history of failure accumulation – accrued

over time scales orders of magnitude much larger than the Maxwell relaxation time – as well as a time-dependent forcing by the ocean on the floating ice (e.g., Bindschadler et al., 2011). For example, calving via the detachment of large, tabular icebergs is an important end-member of observed calving styles (Amundsen and Truffer, 2010). The fractures that determine the size of very large icebergs—such as the 'loose tooth' at the terminus of the Amery Ice Shelf, and thus the calving rate in these locations – are exactly those features that result from ductile and brittle deformation over yearly to decadal time scales (Bassis et al., 2008). Thus while the elastic component of stress relaxes away over long-term simulations, the fractures resulting from the elastic component of stress remain and affect the ice dynamics (e.g., the complete disintegration of the Larsen B Ice Shelf, examined by Glasser and Scambos, 2008).

   In this paper we employ a Lagrangian finite element model with explicit time integration that allows for both the elastic and viscous components of ice deformation to be taken in to account, while simulating ice failure on unstructured meshes. We examine how failure zones form and propagate in an advecting ice slab as it loses contact with the underlying bedrock and begins to float while relaxing the rheological assumption of ice as a purely non-linear viscous material. The model exploration here is not meant to be wide-reaching and exhaustive; rather, it is presented as a tool to aid in the exploration of how the failure of ice impacts its flow, and how rheological assumptions result in different qualitative expressions of ice flow and failure.

## 2 Model description

DES (DynEarthSol3D) is a robust, adaptive, two- and three-dimensional finite element model that solves the momentum balance and heat equation in Lagrangian form using unstructured meshes.

### 2.1 Equations of motion

While many FS models neglect acceleration and formulate ice flow as a static problem, momentum conservation in DES takes the full dynamic form:

$$\rho\dot{u} = \nabla \cdot \sigma + \rho g, \tag{2}$$

where $\rho$ is the material density, $\boldsymbol{u}$ is the velocity vector, $\boldsymbol{\sigma}$ is the Cauchy stress tensor, and $\boldsymbol{g}$ is the acceleration due to gravity. The dot above $\boldsymbol{u}$ is the total time derivative, and variables in boldface are vectors or tensors. The $\nabla\cdot$ is the divergence operator. DES is designed to solve dynamic and quasi-static problems by applying the dynamic relaxation technique (Cundall, 1989) to Eq. (2), of which details are given below. When solving the FS with incompressible fluid assumption, the pressure and velocity are decoupled. Hence, pressure becomes an independent variable. DES does not assume ice is an imcompressible fluid (this is contrary to FS / SIA / SSA ice flow models). The pressure is derived from the elastic deformation of volume change and is not an independent variable, and we do not enforce mass conservation in the typical way that FS (or SIA / SSA) models do.

The temperature field of the ice is modeled using the following heat equation:

$$\rho c_p \dot{T} + \boldsymbol{v} \cdot \nabla \boldsymbol{T} = k\nabla^2 \boldsymbol{T}, \tag{3}$$

where $T$ is the temperature in Kelvin, $c_p$ is the heat capacity of ice, and $k$ is the thermal conductivity. We do not include the effects of deformational strain heating within the ice. For the temperature field we impose Dirichlet boundary conditions on the ice surface and base, as well as at any water boundaries. Often in numerical ice models heat fluxes are applied at the boundary. However, because Neumann conditions are not yet accommodated by DES, we cannot apply basal heat fluxes and instead prescribe Dirichlet boundary conditions in temperature. Further, while we do not evolve the boundary temperature throughout time, the ice temperature does impact the nonlinear viscosity, and so we allow this factor to be present in our simulations.

The governing equations are discretized using an unstructured mesh composed of triangular (2D) or tetrahedral (3D) elements. The approximate displacement $\boldsymbol{x}$, velocity $\boldsymbol{u}$, acceleration $\boldsymbol{a}$, force $\boldsymbol{f}$, and temperature $T$ are defined with linear basis functions (i.e., with P1 elements), while other physical quantities (e.g., stress $\boldsymbol{\sigma}$ and strain $\boldsymbol{\varepsilon}$) and material properties (e.g., density $\rho$ and viscosity $\eta$) are piecewise constant over elements. Conservation of mass is enforced via elasticity rather than the incompressibility condition. A general schematic of DES' solution scheme is shown in Fig. 1.

In DES we make use of both stress and velocity Dirichlet boundary conditions. Subaerial ice is subject to a traction-free boundary condition, or $\boldsymbol{\sigma} \cdot \boldsymbol{n} = \boldsymbol{0}$. Floating ice is subject to an applied normal stress equal to the weight of the water column displaced by the ice plus an additional, diurnal tidal signal of 1 m amplitude. As yet, the location of grounding lines are prescribed a priori in DES and do not evolve according to ice thickness or other environmental variables.

## 2.2 Constitutive relations

The following is a recapitulation of the presentation of DES' constitutive formulation presented by Choi et al. [2013]. The interested reader is directed to that work and its references that describe the well-established field of the numerical modeling of large-strain continuum problems involving material failure and elasticity. DES can accumulate large strains by adding up small-strain increments, and the use of small strains at every time step is justified by the relatively small time step size.

The user has the choice in DES of either evaluating the stress field as linear elastic (with the option of employing a Mohr-Coulomb failure threshold), or linear Maxwell viscoelastic (with no associated failure threshold). We use the former (often termed elastoplastic) to approximate the brittle, rupture-prone behavior of ice. In tectonophysics, for example, elastoplasticity is often understood as the formation or activation of faults. We take a similar interpretation of this rheology as applied to ice: while in our experiments ice is initially rupture-free, the brittle failure of ice approximated by this rheology indicates both the initiation of zones of failure and enables the re-activation of older zones of failure.

For the ductile, rupture-free behavior of ice we use Maxwell viscoelasticity with no associated failure threshold. Finally, as ice in nature both simultaneously flows and ruptures, we combine the two stress evaluations in a third constitutive framework which we call semi-brittle. This case simply calculates both the brittle (elastoplastic) and ductile (viscoelastic) stresses at each point in time in the domain and, depending on a strain-rate threshold (Schulson and Duval, 2009), selects the ductile stress

if the local strain rate is below the threshold and the brittle stress if the strain rate is above the threshold. Many numerical and material parameters play a role in these stress evaluations, and we have tuned those parameters to match laboratory-derived strain- and strain-rate-versus-time curves (Appendix A).

In DES the updated stress tensor in the momentum equation is calculated using the strain rate and strain tensors. These are determined by the constitutive relation. For the ductile (Maxwell viscoelastic) rheology, viscosity is determined by Glen's flow law:

$$\eta = \frac{1}{2} A^{-1/n} \dot{\varepsilon}_e^{(1-n)/n}. \tag{4}$$

The stress update is given by:

$$dev(\boldsymbol{\sigma}^{t+\Delta t}) = 2\eta \, dev(\dot{\boldsymbol{\varepsilon}}^t) + K \, tr(\boldsymbol{\varepsilon}), \tag{5}$$

where dev(*) and tr(*) indicate the deviatoric component and trace of the quantity in the parentheses, and $K$ is the bulk modulus. For the ductile (viscoelastic, VE) rheology the constitutive update is given by the total deviatoric strain increment which is composed of a viscous and elastic contribution corresponding to the mechanical analog of a spring and dashpot in series (a Maxwell element):

$$dev(\Delta\boldsymbol{\varepsilon}) = \frac{dev(\Delta\boldsymbol{\sigma}_{VE})}{2G} + \frac{dev(\boldsymbol{\sigma}_{VE})\,\Delta t}{2\eta}. \tag{6}$$

Substituting $\Delta\boldsymbol{\varepsilon}$ with $\boldsymbol{\varepsilon}^{t+\Delta t} - \boldsymbol{\varepsilon}^t$, $\Delta\boldsymbol{\sigma}_{VE}$ with $\boldsymbol{\sigma}_{VE}^{t+\Delta t} - \boldsymbol{\sigma}^t$, and $\boldsymbol{\sigma}_{VE}$ with $(\boldsymbol{\sigma}_{VE}^{t+\Delta t} + \boldsymbol{\sigma}^t)/2$, the equation above is reduced to:

$$dev(\boldsymbol{\sigma}_{VE}^{t+\Delta t}) = \frac{\left(1 - \frac{G\Delta t}{2\eta}\right)dev(\boldsymbol{\sigma}^t) + 2G \cdot dev(\boldsymbol{\varepsilon}^{t+\Delta t} - \boldsymbol{\varepsilon}^t)}{1 + \frac{G\Delta t}{2\eta}}. \tag{7}$$

The isotropic stress components are updated based on the volume change. As a results, the viscoelastic stress is the following:

$$\boldsymbol{\sigma}_{VE}^{t+\Delta t} = dev(\boldsymbol{\sigma}_{VE}^{t+\Delta t}) + \Delta t \, K \, tr(\dot{\boldsymbol{\varepsilon}}^{t+\Delta t})\,\boldsymbol{I}. \tag{8}$$

The brittle (elastoplastic, EP) stress $\boldsymbol{\sigma}_{EP}$ is computed using linear elasticity and the Mohr-Coulomb (MC) failure criterion with a general (associative or non-associative) flow rule. Following a standard operator-splitting scheme (e.g., Lubliner, 1990; Simo and Hughes, 2004; Wilkins, 1964), an elastic trial stress $\boldsymbol{\sigma}_{et}^{t+\ \Delta t}$ is first calculated as:

$$\boldsymbol{\sigma}_{et}^{t+\ \Delta t} = \boldsymbol{\sigma}^t + \left(K - \frac{2}{3}G\right) tr(\dot{\boldsymbol{\varepsilon}}^{t+\ \Delta t})\boldsymbol{I}\Delta t + 2G\dot{\boldsymbol{\varepsilon}}^{t+\ \Delta t}\Delta t, \tag{9}$$

If the elastic trial stress is within a yield surface, that is, $f\left(\boldsymbol{\sigma}_{et}^{t+\ \Delta t}\right) > 0$, where $f$ is the yield function, then the stress does not need a plastic correction. In this case $\boldsymbol{\sigma}_{ep}^{t+\ \Delta t}$ is set equal to $\boldsymbol{\sigma}_{et}^{t+\ \Delta t}$. If $\boldsymbol{\sigma}_{et}^{t+\ \Delta t}$ is on or outside the yield surface, then it is projected onto the yield surface using a return-mapping algorithm, (e.g., Simo and Hughes, 2004).

In the case of a Mohr-Coulomb material, it is convenient to express the yield function for the tensile failure as

$$f_t(\sigma_3) = \sigma_3 - \sigma_t, \tag{10}$$

where $\sigma_3$ is the minimal principal stress (with the convention that tension is positive), and $\sigma_t$ is the yield stress in tension. For shear failure the corresponding stress envelope is defined as

$$f_s(\sigma_1, \sigma_3) = \sigma_1 - N_\phi \sigma_3 + 2C\sqrt{N_\phi}, \tag{11}$$

where $\sigma_1$ is the maximal principal stress, $C$ is the material's cohesion, $\phi$ is the internal friction angle, and $N_\phi = \frac{1+\sin\phi}{1-\sin\phi}$. To guarantee a unique decision on the mode of yielding (tensile versus shear), we define an additional function that bisects the obtuse angle made by two-yield function on the $\sigma_1 - \sigma_3$ plane, as

$$f_h(\sigma_1, \sigma_3) = \sigma_3 - \sigma_t + \left(\sqrt{N_\phi^2 + 1} + N_\phi\right)\left(\sigma_1 - N_\phi \sigma_t + 2C\sqrt{N_\phi}\right) \tag{12}.$$

Once yielding occurs, that is $f_s < 0$ or $f_t > 0$, the mode of failure is decided based on the value of $f_h$. Shear failure occurs if $f_h < 0$ and tensile otherwise. Ice is much stronger in compression than in tension, and, as we herein do no attempt to simulate any ice flowing over a pinning point or other such

obstruction that would favour a compressive stress regime, we do not account for the compressive failure of ice (which would necessitate the employment of a failure threshold with very different properties).

Frictional materials generally follow a non-associative flow rule, meaning the direction of plastic flow in the principal stress space is not the same as the direction of the vector normal to the yield surface. The plastic flow potential for tensile failure can be defined as

$$g_t(\sigma_3) = \sigma_3 - \sigma_t,$$
(13)

while the plastic flow potential for shear is

$$g_s(\sigma_1, \sigma_3) = \sigma_1 - \frac{1+\sin\psi}{1-\sin\psi}\sigma_3$$
(14).

When there is plastic failure, the total strain increment is given by

$$\Delta\boldsymbol{\varepsilon} = \Delta\boldsymbol{\varepsilon}_{el} + \Delta\boldsymbol{\varepsilon}_{pl},$$
(15)

where $\Delta\boldsymbol{\varepsilon}_{el}$ and $\Delta\boldsymbol{\varepsilon}_{pl}$ are the elastic and plastic strain increments. The plastic strain increment is normal to the flow potential surface and can be written as:

$$\Delta\varepsilon_{pl} = \beta\frac{\delta g}{\delta\sigma}$$
(16)

where $\beta$ is the plastic flow magnitude. $\beta$ is computed by requiring that the updated stress state lies on the yield surface,

$$f(\boldsymbol{\sigma}_{ep}^{t+\Delta t}) = f(\boldsymbol{\sigma}^t + \Delta\boldsymbol{\sigma}_{ep}) = 0$$
(17).

In the principal component representation, $\boldsymbol{\sigma}_A = \boldsymbol{E}_{AB}\boldsymbol{\epsilon}_B$ where $\boldsymbol{\sigma}_A$ and $\boldsymbol{\epsilon}_A$ are the principal stress and strain, respectively, and $\boldsymbol{E}$ is a corresponding elastic moduli matrix with the following components:

$$E_{AB} = \left(K - \frac{2}{3}G\right) \text{ if } A \neq B \text{ and}$$
(18.a)

$$E_{AB} = \left(K + \frac{4}{3}G\right) \text{ otherwise}$$
(18.b).

By applying the consistency condition and using $\boldsymbol{\sigma}_{et}^{t+\Delta t} = \boldsymbol{\sigma}^t + \boldsymbol{E}\cdot\Delta\boldsymbol{\epsilon}$, we obtain the following formula for $\beta$,

$$\beta = \frac{\sigma_{el,3}^{t+\Delta t} - \sigma_t}{\frac{\delta g_t}{\delta\sigma_3}} \text{ for tensile failure and}$$
(19.a)

$$\beta = \frac{\sigma_{el,l}^{t+\Delta t} - N_\phi \sigma_{el,3}^{t+\Delta t} + 2C\sqrt{N_\phi}}{\sum_B (E_{1B}\frac{\delta g_s}{\delta \sigma_B} - N_\phi E_{3B}\frac{\delta g_t}{\delta \sigma_B})} \text{ for shear failure}$$ (19.b).

Likewise, $\delta g/\delta \sigma$ takes different forms according to the failure mode:

$$\delta g/\delta \sigma_1 = 0$$ (20.a),

$$\delta g/\delta \sigma_2 = 0$$ (20.b),

$$\delta g/\delta \sigma_3 = 1 \text{ for tensile failure and}$$ (20.c)

$$\delta g/\delta \sigma_1 = 1$$ (21.a)

$$\delta g/\delta \sigma_2 = 0$$ (21.b)

$$\delta g/\delta \sigma_3 = \frac{1+\sin\psi}{1-\sin\psi} \text{ for shear failure.}$$ (21.c)

Once $\Delta \boldsymbol{\varepsilon}_{pl}$ is computed, $\boldsymbol{\sigma}_{ep}$ is updated as

$$\boldsymbol{\sigma}_{ep} = \boldsymbol{\sigma}_{et}^{t+\Delta t} - \boldsymbol{E} \cdot \Delta \boldsymbol{\varepsilon}_{pl}$$ (22)

in the principal component representation and transformed back to the original coordinate system. After the viscoelastic stress and elastoplastic stress are evaluated, we compute the second invariant of the deviatoric components of each, and following the minimum energy principle, select the smaller of the two as that element's stress update.

## 2.3 Numerical considerations

DES is formulated as a finite element method with explicit time integration, and the order of calculations can be seen in Fig. 1. The advantage of using this method is that the computational cost of each time step is small (compared to implicit methods where advancing by one large time step involves the solving of large, ill-conditioned linear systems) and the implementation of non-linear rheologies is simple.

The use of the explicit time integration means that the time step is limited to very small values, on the order of $\Delta X_{min}/u_{elastic}$ where $\Delta X_{min}$ is the smallest edge length of an element and $u_{elastic}$ is the elastic wave speed (from the Courant-Friederics-Lewy condition). We overcome this limitation using

the mass scaling technique that is detailed in Choi et al. (2013). Because DES employs a suite of constitutive relations, we also need to consider the constraints on the time step size associated with the dominating deformational mechanism. The time step in these simulations is chosen as the minimum between

5 $$\Delta t = \min\{\,\Delta t_{elastic}\,,\Delta t_{maxwell}\,\} \tag{23}$$

where

$$\Delta t_{elastic} = \Delta X_{min}/2c\,u_{char} \tag{24.a}$$

$$\Delta t_{maxwell} = \eta_{min}/4G \tag{24.b}$$

where $c$ is the inertial scaling parameter related to the dynamic relaxation, $\Delta u_{char}$ is the characteristic 10 advective speed, $\eta_{min}$ is the minimum allowable viscosity, and $G$ is the shear modulus. This scheme ensures that the dominating deformational mechanism is adequately resolved in time. As such the time steps in DES are on the order of seconds to hours, depending largely on mesh parameters and the characteristic speed of the simulation as determined by the phenomenon the user wishes to resolve. Typical values for $c$ have been found to fall in the range of $\{10^4, 10^8\}$ (Choi et al., 2013) and we find 15 that a value of $c = 10^5$ works well for semi-brittle ice (see Appendix A and B for ice calibration and benchmark).

In addition to the dynamic time-stepping routine, several other numerical techniques are employed that distinguish this model from implicit finite element schemes commonly used to solve FS systems. DES solves the dynamic momentum balance equation, Eq. (2), by damping the inertial forces 20 at each time step, giving rise to the quasi-static (i.e., static with time-dependent boundary conditions) solution. Originally proposed by Cundall [1989], this variant of dynamic relaxation applies forces at each node in the domain opposing the direction of the node's velocity vector:

$$m\boldsymbol{a}_i = (\boldsymbol{f}_{damped})_i = \boldsymbol{f}_i - \chi\,sgn\,(\boldsymbol{u}_i)|\boldsymbol{f}_i| \tag{25}$$

where the subscript $i$ denotes the $i$-th component of a vector and the $sgn(*)$ denotes the signum function. 25 $\chi$ is a user-supplied damping factor ($\chi = 0.8$ has been shown to ensure stability, e.g., Choi et al., 2013). This numerical method is employed often when simulating complex rheologies (typically encountered

in tectonophysics, e.g., Wu and Lavier, 2016) as a means of obviating the need to solve large, non-linear sets of equations encountered in purely Stokes formulations of ice flow problems.

The linear triangular elements used in DES are known to suffer volumetric locking when subject to incompressible deformations (e.g., Hughes, 2000). Because we model phenomena that require incompressible plastic and viscous flow, we use an anti-volumetric-locking correction based on the nodal mixed discretization methodology (Detournay and Dzik, 2006; De Micheli and Mocellin, 2009). The technique simply averages the volumetric strain rate over a group of neighboring elements and then replaces each element's volumetric strain rate with the averaged one. Choi et al. [2013] describes this technique in greater detail.

Finally, DES makes use of adaptive remeshing. Based on the quality constraints selected by the user, DES assesses the mesh quality at fixed step intervals and remeshes if elements are found in violation (e.g., if a triangular element contains an angle smaller than some input threshold). New nodes may be inserted into the mesh (or old ones deleted) and the mesh topology can be changed through edge flipping. The nodes are provided to the Triangle library (Shewchuk, 1996) to construct a new triangulation of the domain. After the new mesh is created, the boundary conditions, derivatives of shape functions, and mass matrix are recalculated. When deformation is distributed over a large region or the whole domain, remeshing may result in a new mesh quite different from the old one. Because of this possibility the fields associated with nodes (e.g., velocity and temperature) are linearly interpolated from the old mesh to the new. For data associated with elements (e.g., strain and stress) DES uses an approximate conservative mapping described in detail by Ta et al. [2015].

**3 Experiments: different constitutive models for ice**

Observations have shown that the bending that occurs as ice transitions from resting on land to floating in water (the grounding line) promotes the failure of ice from the bottom up, called basal crevasses, that often appear with characteristic regularity in spacing, persisting within the ice for long distances and eventually promoting the calving of ice (Bindschadler et al., 2011; Glasser and Scambos, 2008; Logan et al., 2013; McGrath et al., 2012; James et al., 2014; Murray et al., 2014). The main motivation of

using DES to understand this phenomenon (or a simplified version thereof) is its rheological flexibility. That is, a wide array of phenomena in nature may be explored in DES by relaxing the assumption that ice is a purely non-linear viscous fluid, and examining how ice deformation differs if ice is assumed to be ductile, brittle, or some mixture of the two.

To distill the effects that certain constitutive choices have on the time-dependent ice deformation, we divide this section according to how different constitutive models available within DES simulate the ductile, brittle, and semi-brittle deformation of ice in two different geometrically simple experiments: Experiment 1, a tilted, planar, pseudo-rigid box being advected through a bending fulcrum (Fig. 2a); and Experiment 2, a flat wedge undergoing a transition from frozen or freely-slipping

to buoyantly floating with an added 1 m diurnal tidal signal (Fig. 2b). Both experiments are initialized with the same temperature configuration, and the temperatures on the boundaries are fixed throughout the experiments. These experiments, while extremely simple, are designed in order to examine the effect that bending has on these relatively complicated rheologies. They are not meant as realistic, glacier-like scenarios, but rather are idealized scenarios designed solely for the purpose of

understanding the range of deformation behaviours for different constitutive formulations. Their purposeful simplicity allows us to examine the effect that bending has on ice and to attribute deformation and flow patterns solely to the choice in rheology.

### 3.1 Purely ductile or brittle ice

For Experiment 1, we set the ice thickness to 1000 m and prescribe the inflow, basal, and
outflow side velocities to be 300 m yr$^{-1}$, as these are realistic values for marine outlet glaciers. While basal boundary conditions in glaciers are often either formulated as a sliding law or completely frozen / free-slip, we have chosen to prescribe Dirichlet boundary to ensure the ice slab advects through a bending fulcrum (results from Experiment 2 will show this is necessary for purely brittle ice). The initial mesh resolution is 50 m. The inflow and basal boundary velocities are prescribed at an angle of 3

degrees until they reach 10 km, at which point they are forced horizontal. The temperature of the ice is defined by a linear gradient between Dirichlet conditions of −30° C at the surface to 0° C at the base.

Figure 3 shows the effective stress (a), strain rate (c) and viscosity (e) after twenty years model time for purely brittle ice. Purely brittle ice experiences stress 1 to 2 orders of magnitude higher than purely ductile ice (Figs. 3a and b). For the brittle case, stresses are highest (about 1 MPa) on the basal surface and in a thin vertical line where the surface bends at the grounding line, whereas in the ductile ice case we observe only a slight increase in stress at the grounding line and nowhere else. Overall we observe in both experiments lower viscosity ice at the grounding line and higher ice viscosity upstream and downstream of the grounding line (Figs. 3e and f), however for the ductile case there is a much wider zone of low viscosity ice – by as much as 2 orders of magnitude smaller than the brittle ice. This means that the strains associated with deformation are more localized for the brittle rheology (limited to about 1 or 2 elements in width making for a very small deformational process zone) and conversely very diffuse for the ductile rheology. The comparative weakness and low effective viscosity of the ductile rheology is reflected by the surface topography: the left-hand side of the domain in the ductile simulation shows a depression at the ice surface, where the ice is essentially slumping toward the right-hand side of the domain under the force of gravity. Additionally, the velocity boundary condition imposed at the left side of the domain introduces an artifact in the flow that is expressed as an artificial steep surface depression: while the boundary nodes' velocities are prescribed, the nearby interior nodes relax and flow downhill under the force of gravity.

Because these simulations are intended to only compare ductile and brittle approximations for ice flow, the ductile ice does not fail (no yield envelope has been provided for this stress calculation). The brittle ice can and does fail, however, as dictated by the Mohr-Coulomb threshold, with a pattern shown in Fig. 4. Reasonable values for the yield envelope properties were selected from the literature and are listed in Table 1 (Bassis and Jacobs, 2013; Fish and Zaretsky, 1997; Sammonds et al., 1998). The plastic strain (or amount of strain a failed element undergoes once it has reached yield stress) for the brittle ice shows a very regular, localized pattern. We executed the same experiments with applied velocities of 600 and 900 m yr$^{-1}$ and saw no difference in the spacing or amount of strain. That is, failure patterns were insensitive to the speed at which the slab was advected down the slope and through the bend. During model simulation the strain begins at the base of the slab at the bending fulcrum and quickly propagates upward toward the surface. Ice upstream of the bending fulcrum remains fully intact

until reaching the thin process zone delineated by the stress and strain rate fields shown in Fig. 3a and c. Once the ice has been advected away from this thin zone of high stress and strain rate the accumulation of post-failure strain ceases, leaving a pattern of regularly spaced, thin, vertical lines of failed ice, that have failed in sequence.

To ensure that the kinematic velocity conditions are not contaminating the stress field in Experiment 1 (Fig. 2a), we executed the purely brittle and ductile ice according to the setup for Experiment 2 in Fig. 2b, with a flat, freely slipping or frozen bed. We maintain a static grounding line, and due to numerical constraints on DES' remeshing algorithm, we cannot maintain the initial thickness gradient; thus, the driving stress decreases throughout the simulation, leading to a model time of

approximately 3 years. The geometry of the domain is shown in Fig. 2b, where the thickness of the left side is 1050 m, decreasing linearly over 50 km to 900 m on the right. We found that this initial thickness gradient produces a driving stress with reasonable terminus velocities matching to those of glaciers with ice shelves in Antarctica (Rignot et al., 2011). We employed the geometric setup in Experiment 1 because purely brittle ice initialized as shown in Fig. 2b does not flow; it remains static

(Fig. 5a and c). Figure 5 shows the effective stress and viscosity of brittle and ductile ice after 1 year of model time. Again, the brittle ice experiences stresses an order of magnitude higher than the ductile ice (Fig. 5a and b) and a much smaller process zone of high stress at the grounding line, and the resulting viscosity field after 1 year model time (Fig. 5c and d) shows a large difference: 2 orders of magnitude at the grounding line and as much as 4 or 5 upstream. The brittle ice remains essentially in its initial

configuration at the end of 1 year, having accumulated a maximum of $10^{-3}$ effective strain in its most deformed element, while the ductile ice has advanced 2700 m past its initial location, having accumulated $10^{-1}$ effective strain in its most deformed element. For both cases, a frozen bed results in almost no movement after 10 years model time.

**3.2 Semi-brittle behaviour**

Here we investigate the partitioning of viscous or ductile ice flow and brittle failure under boundary conditions that promote the formation of basal crevasses at glacier grounding lines – areas of fast ice movement and flexure. We suggest this may represent an advance from previous damage-centric

models where damage is estimated either in static snapshots throughout time (Borstad et al., 2016) for entire ice sheets or in completely time-dependent but small-strain conditions, as in Duddu et al., 2013 (i.e., the domain was not characterized by strains exceeding 100 % with advecting ice).

In glaciological literature there is evidence for a transition from ductile flow to brittle failure depending on the applied strain rate: specimens of ice experiencing low strain rate flow in a ductile manner, with viscosities adhering to Eq. 7, and those straining faster than a laboratory observed value of $10^{-7}$ s$^{-1}$ fail in a brittle manner (Schulson and Duval, 2009, chapter 9). Simply following these observations: DES selects either the ductile or brittle constitutive update based on the local strain rate field (see the steps in the pseudo-code, Fig. 1). Elements in the domain with a strain rate less than $10^{-7}$ s$^{-1}$ are approximated as ductile (or Maxwell viscoelastic) and elements straining faster are approximated as brittle (or Mohr-Coulomb elastoplastic). The semi-brittle rheology employed here is supported by laboratory data that show that a stiffening of ice at a high strain rate will be accompanied by fracture only at correspondingly high tensile stress (from $10^5$ to $10^6$ Pa) (Bassis and Jacobs, 2013; Schulson and Duval, 2009). The strain rate dependent nature of the transition from ductile to brittle implies that – depending on the viscosity – fracture in ice occurs on time scales of less than a few seconds to hours, which DES easily resolves. Appendix A presents the weakening parameters used to calibrate semi-brittle ice in DES against laboratory-derived strain- and strain-rate- versus-time curves, as well as lists all relevant material and numerical parameters that reproduce this behavior (following essentially the same exercise in Duddu and Waisman, 2012). Most important, we recover the value of plastic strain that indicates when semi-brittle ice has ruptured. Appendix B shows the results of DES executed according to Experiment E (Haut Glacier d'Arolla) from the ISMIP-HOM intercomparison project with the values obtained from the calibration experiments, to show that semi-brittle ice largely reproduces the behavior in this benchmark test.

We execute DES with semi-brittle ice according to the setup in Experiment 2 for 2 different mesh sizes, 100 and 50 m (computational resources did not allow for 25 m resolution). No ice melting is applied to these boundaries as this effect is the subject of future work.

Figure 6 shows the velocity, effective stress, strain rate, and viscosity at 6 months model time. Up until this time in the simulation the ice reaches a maximum velocity of about 2 km a$^{-1}$, after which the velocities decrease to 0 due to a loss of driving stress as the ice extends into the floating portion of the domain. Stresses at the grounding line in these simulations are high: about 1 MPa at the grounding line, similar to the behavior seen in previous experiments (Fig. 3).

We also determine the distribution of ice failure for the semi-brittle rheology (Fig. 7). Ice at the surface is regularly and heavily broken as the yield strength there is the lowest (this is the case for all frictional materials in the vertical plane). As the floating portion of the ice extends further past the grounding line the ice thins, allowing for necking at the grounding line and at other places the ice has failed in the floating tongue. This thinning as ice begins to float is a feature of marine-terminating ice sheets, and is accentuated in nature by intense basal melting. Toward the end of the simulation the floating tongue has accumulated so much strain that it begins to form undulated pinch-and-swell structures (Fig. 7). We term this characteristic pinch-and-swell geometry boudins, where we count 18 boudins that have a mean spacing of 530 m and a standard deviation of 150 m at the end of the 3 year model run. While the basal crevasses form sequentially, that is – failed ice to the right of the domain are older than those to the left – these features develop more fully into the characteristic boudin-like shape all at the same time in the model. Once the ice has lost all its driving stress the ice begins to thicken just beyond the grounding line, which is a consequence of the boundary conditions and the lack of true bedrock below the ice. As in the purely brittle ice in Experiment 2, simulations with a frozen bed resulted in almost no deformation at all.

Computational limitations prevented the simulation of Experiment 2 under quartered resolution, and so Fig. 8 shows the result of the same semi-brittle, freely slipping ice after 8 months model time for halved resolution. Boudins develop again here, although at much shorter wavelength than for a resolution of 100 m: they have a mean spacing of 250 m with a standard deviation of 83 m, approximately half that of the coarser experiment. Gone however is the rather jagged, undulating surface shown in Fig. 8a; instead, Fig. 8b shows that mesh refinement produces a much smoother surface in ice that was initially floating in the domain (and did not traverse the grounding line). Experiments with 100 m resolution took less than 1 hour to complete, and those with 50 m resolution

completed in approximately 24 hours. Lastly, to ensure that the unrealistic initial geometry of Experiment 2 does not contaminate results, we ran experiments where ice was held in its initial geometry before flowing out across the grounding line, to allow the effect of elastic shocks to decay. Model results were the same for both coarser and finer resolution: boudins formed with the same regularity and pattern.

## 4 Discussion and conclusion

The experiments performed in this study are not meant to be exhaustive and wide-reaching; rather, they were performed to show how a semi-brittle ice-like material responds to very idealized initial and boundary conditions. Because we do not actually simulate fractures – ice in DES is represented as a continuum material – we must assume that at some level of plastic strain, the ice in a simulation is considered broken. Appendix A shows calibration experiments wherein we determined that an accumulated plastic strain value of 0.03 is sufficient to consider semi-brittle ice to have ruptured. Zones of intense, vertical localization in these experiments can be considered to have ruptured for plastic strain values > 0.03 and further, in essence, could represent basal crevasses. These 'crevasses' initiate in virgin material where the shear and extensional stresses due to bending and increases in velocity are highest and are then advected downstream from the grounding line. While Experiment 1 is very simple, the regularly-spaced zones of failure motivate the use of a semi-brittle material, which is tested under more realistic conditions in Experiment 2. One perhaps counter-intuitive result of Experiment 1 is that the brittle stresses are much higher (about an order of magnitude) than the ductile stresses. In essence, this difference stems from the fact that in viscoelasticity the elastic stresses are continually relaxed, while they remain in elastoplasticity. Equations 8 and 9 provide explicit expressions that show for all strain rates in these experiments, the given bulk and shear moduli, and the viscous parameters used, that the elastic (brittle) stresses are always higher than the viscoelastic (ductile) stresses. The initial and boundary conditions of Experiment 2 represent a next step toward a more realistic simulation as the ice flow is generated entirely by a gradient in ice thickness and we employ a jump in boundary conditions (from freely slipping or frozen to floating) at the grounding line. As this work represents an initial exploration and presentation of the model's capability, simulations that employ a sliding law and the

effects of accumulation and ablation are left for later studies. For example, the loss of driving stress in the semi-brittle experiment could be mitigated by applying a velocity condition on the inflow boundary, or simply by remeshing the upstream side geometry to its original shape. Our focus here is on the effect of rheological choices, and with this goal in mind we have attempted to design experiments idealized

enough so that differences in deformation can be attributed to rheology. Similarly, while our simulations are carried out under the condition of conservation of energy, we do not explore the effect of changing ice temperature on ice flow; rather, we carry out the experiments under this condition to demonstrate that future experiments will be able to solve both conservation of momentum (with non-zero acceleration) and energy without too much computational cost. Simulations presented here ranged

in their computational cost, from less than one hour to less than one day. Certainly, experiments with even finer resolution are left to future work as the computational cost to perform them becomes much higher.

One main result from these experiments is the observation that boudins may form as a consequence of semi-brittle rheology. This observation may appear to be complicated by the fact that

the boudin size or spacing scale with mesh size. Failure in ice is marked by localized strain, and computational strain localization is well-known to be mesh-dependent under rate-independent plasticity, which is the brittle rheology implemented in DES. In this sense then ice failure in DES scales with element size, and this is consistent with and predicted by rate-independent plasticity. While it may be unsatisfying to the reader that these geometric feature's sizes are sensitive to resolution, we emphasize

that they are a qualitative (rather than quantitative) observation, resulting from this specific semi-brittle rheology and jump in boundary condition from slipping to floating. Future work needs to be devoted to examining the convergence of this feature against increased resolution. Our results however would be inconsistent with those developed in a theoretical perturbation formalism by Bassis and Ma (2015), wherein the dominant wavelength for boudin spacing was on the order of ice thickness (our experiments

here show wavelengths of approximately half to quarter ice thickness). These differences might reflect the primary assumptions underpinning the two approaches – one is viscous (although allows for a brittle limit) and the other is semi-brittle. Further, the formulation developed by Bassis and Ma (2015) permits a central role for basal melting within basal crevasses, an undoubtedly crucial feature that DES does not

implement in a sophisticated way at present. Care should be taken in extrapolating results here to real glaciers; these experiments are performed only as an initial exploration of the potential for this kind of rheological framework to aid in understanding patterns of flow and failure seen in nature. Certainly, that DES lacks the implementation of Neumann boundary conditions indicates that studies where constant ice fluxes must be maintained are best left to other numerical models at the moment. More work must also be conducted to understand the competing effects of viscoelastic damping and brittle failure propagation.

From very simple model runs we learned there may indeed be a ductile to brittle transition in ice that is likely very difficult to capture in many numerical models. Figure 9 indicates how we imagine failure and subsequent deformation occurring in floating ice masses in nature: at grounding lines, both an increase in ice velocity (due to loss of retaining frictional forces applied by bedrock contact) and an application of bending moment (due to tides and the equilibrated response beams and plates exhibit when they are partially supported by fluid) lead to high stresses and strain rates that initiate ice failure from the bottom up (Fig. 9a). As ice accelerates into open ocean it thins, promoting further crack propagation, which can be further widened by intrusions of warm, buoyant melt water (Fig. 9b – not explored in these experiments). Further, thinning, stretching, and ice melting when simulated with a semi-brittle rheology like the one presented here can lead to ice geometries that are like those seen in nature (Bindschadler et al., 2011; Luckman et al., 2012). Since the location and size of basal crevasses can directly impact calving rates by propagating upward through the full thickness of the ice (Logan et al., 2013) understanding their evolution and growth may be critical to predicting calving occurrence and terminus position.

Future work with DES must explore the utility of a semi-brittle ice rheology in more realistic scenarios and with the inclusion of a freely varying grounding line (e.g., one that evolves based on ice thickness) and basal melting – two ice dynamic processes incorporated in other numerical models and known to be critical processes in glacier and ice sheet retreat. At present this study has shown that the assumption of a semi-brittle ice rheology can reproduce the brittle rupture of ice, general ice flow characteristics, and idealized patterns of failure in simple situations, and may be recommended as a tool through which future studies of ice failure related to calving and ice dynamics can be conducted.

**Appendix A Semi-brittle ice calibration**

In an effort to ensure that flow and deformation as represented in DES is reasonable, we calibrated the numerical and material parameters of semi-brittle ice to match strain- and strain-rate-versus-time experiments performed on laboratory-derived ice (Fig. 10). We essentially followed the same exercise as in Duddu and Waisman (2012), in which material parameters are calibrated against deformation curves derived by Mahrenholtz and Wu (1992). While this exercise has its limitations (laboratory-grown ice does not necessarily represent natural ice), it is our attempt at reproducing ice behaviour to the best of DES' ability. Ice in both the laboratory setting and DES was isothermal at -10 C and deformed in uniaxial tension. Semi-brittle ice in DES is initialized in a completely undamaged state. The elastic and shear moduli are only slightly temperature dependent (Schulson and Duval, 2009) and we neglect this temperature dependence because their contribution to total strain is negligible during tertiary creep stages which dominate the length of the experiments, and further because the laboratory experiments were isothermal. We have not calibrated the model for compressional strain- and strain-rate-versus-time as our interest is in simulating scenarios where extensional stresses are dominant, as is typically assumed when ice is calving or rifting. The model domain is set to the same geometry as that of the lab experiments, and ice is subjected to 3 different stresses: 0.93, 0.82, and 0.64 MPa. Figures 10a and b show the initial mesh and final rupture of the semi-brittle ice for an applied stress of 0.82 MPa. Figure 10b shows the rupture of semi-brittle ice after approximately 150 hours: from this we see that the accumulated plastic strain (overlaid in grey) is > 0.03 for the ice plug to have ruptured. This represents a threshold above which we can consider semi-brittle ice in DES to have failed. Figures 10c and d show the weakening that the cohesion and angle of internal friction that reproduce the strain- and strain-rate-versus-time curves (Figs. 10e and f) produced by Mahrenholtz and Wu (1992) for the 3 different applied stresses. The material weakens according to the local amount of plastic strain: values of cohesion and friction larger than those shown in Fig. 10c and d fail to produce any rupture, while those below rupture too fast. Fish and Zaretsky [1997] reported cohesion and internal friction values for ice in compression experiments; while the cohesion values they reported are much larger than those suggested here (ice is well known to be stronger in compression than in tension) we find our internal friction angles to be within theirs. Experiments were also performed for ductile-to-brittle strain-rate thresholds of $10^{-8}$ and

$10^{-6}\,s^{-1}$, with similar findings: the lower strain-rate threshold produced rupture too fast, and the higher none at all.

## Appendix B Benchmark: Haut Glacier d'Arolla

In Choi et al. (2013), DES performed the benchmark tests for a range of material or rheological behaviors to validate and verify this numerical method. These tests included: 1 – flexure of a finite-length elastic plate; 2 – thermal diffusion of a half-space cooling plate; 3 – stress build-up in a Maxwell viscoelastic material; 4 – Rayleigh-Taylor instability; and 5 – Mohr-Coulomb oedometer test. Thus DES has been verified and validated and is already in use in fields relating to crustal deformation (Ta et al., 2015). Despite this prior exercise in verification and validation demonstrating that DES' numerics are well understood, we executed DES according to a benchmark test presented by Pattyn et al. (2008) in the ISMIP-HOM study in the spirit of presenting DES as a numerical model suitable for the community of glaciologists. All experiments in Pattyn et al. (2008) were designed to be isothermal and many employ boundary conditions that DES unfortunately cannot accommodate due to its entirely mobile mesh (e.g., periodic boundary conditions). However, Experiment E (Haut Glacier d'Arolla) calls for boundary conditions that DES can easily employ, in 2 tests: first, a completely frozen bed everywhere in the domain, and second, a completely frozen bed except for $2200 \leq x \leq 2500$ m in the domain, where ice slips freely. The flow-law rate factor is set to $A = 10^{-16}\,Pa^{-n}\,a^{-1}$, and the resolution is suggested to be 100 m. DES performed both tests with 10 m resolution, and the results are shown in Fig. 11. Given that DES has a completely different constitutive framework than the FS models that participated in the ISMIP-HOM suite of experiments, the model does remarkably well with small misfit compared to the suite of FS models that participated in the exercise, and within or close to the standard deviation of those models.

**Author contributions**

L. Logan, L. Lavier, and G. Catania helped design the experiments performed herein. L. Logan coded and executed these experiments, and prepared this manuscript. E. Choi and E. Tan developed the code in large part, which was modified by L. Logan and L. Lavier for the experiments in this paper.

**Acknowledgements**

This work was funded by NSF grant ARC-0941678 and the King Abdullah University of Science and Technology. The ice modeling was performed at the University of Texas, Institute for Geophysics; the University of Memphis; and Academia Sinica in Taiwan. The authors gratefully acknowledge A. Vieli, one anonymous reviewer, and Jeremy Bassis for extremely helpful comments that improved this paper.

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

| Symbol | Constant | Value | Units |
|---|---|---|---|
| $\rho$ | Density of ice | 911 | kg m$^{-3}$ |
| $n$ | Power in Glen's Law | 3 | - |
| $A$ | Multiplier in Paterson and Budd (1982) | | |
| | if T < 263 K | $3.615 \times 10^{-13}$ | s$^{-1}$ Pa$^{-3}$ |
| | if T ≥ 263 K | $1.733 \times 10^{3}$ | s$^{-1}$ Pa$^{-3}$ |
| $Q$ | Activation energy for creep in Paterson and Budd (1982) | | |
| | if T < 263 | $6 \times 10^{4}$ | J mol$^{-1}$ |
| | if T ≥ 263 K | $13.9 \times 10^{4}$ | J mol$^{-1}$ |
| $\sigma_T$ | Strength in tension | 1 | MPa |
| $K$ | Bulk modulus | 9500 | MPa |
| $c_p$ | Heat capacity | 2000 | J kg$^{-1}$ K$^{-1}$ |
| $k$ | Thermal conductivity | 2.1 | W m$^{-1}$ K$^{-1}$ |
| $\dot{\varepsilon}_{transition}$ | Ductile-to-brittle transition threshold | $1 \times 10^{-7}$ | s$^{-1}$ |
| $u_{char}$ | Characteristic speed | $1 \times 10^{-6}$ | m s$^{-1}$ |
| $c$ | Inertial scaling | $2 \times 10^{5}$ | - |
| $\chi$ | Inertial damping | .8 | - |

Table 1: Numerical and material parameters used in model runs whose values remained constant.

```
INITIALIZE. Supply list of boundary nodes, areas, holes, and facets to mesh library
FOR each time step n

        Evaluate force at ith node                                              ΣF_i = m_i a_i
        Get new velocities and displacements                                    v_i = dt*a_i ,
                                                                                 x_i = dt*v_i ,
        Evalutate effective strain rate at each element                         E_II = sqrt( ε̇_ij ε̇_ij )

        IF E_II < threshold
                Stress is ductile        := Maxwell viscoelastic constitutive law       σ_D = F( ε, ε̇, T, ... )
        Else
                Stress is brittle        := Mohr-Coulomb elastoplastic constitutive law  σ_B = G( ε, ε̇, T, ... )

        Evaluate new forces from stress and local element area                  dF_i = 3 * σ_e dΩ_e

        Evaluate mesh quality and remesh as necessary
END
```

**Figure 1: Schematic of one time step in DES.**

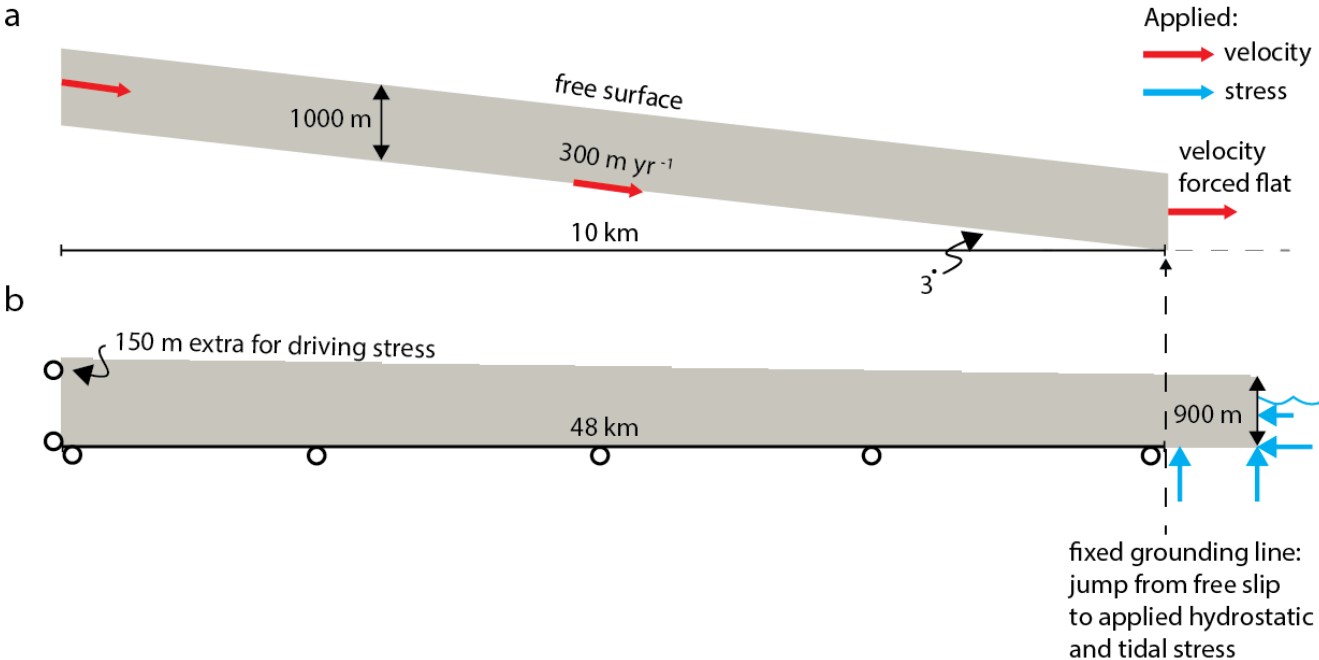

5  **Figure 2: Schematic of experiments. (a) Experiments performed with a purely ductile or brittle rheology are initialized as a parallel-sided slab that is advected down a plane and forced flat at 10 km. (b) Setup for semi-brittle rheology: a horizontal domain of 50 km and a fixed grounding line at 48 km, with an initial 2 km of floating tongue. The left-hand side horizontal velocity is fixed at zero and the bottom side upstream (left) of the grounding line vertical velocity is fixed at zero (this type of boundary condition is represented schematically by open black circles). Hyrdostatic and tidal stress is applied to the bottom and right-hand side of the**
10  **domain downstream (right) of the grounding line.**

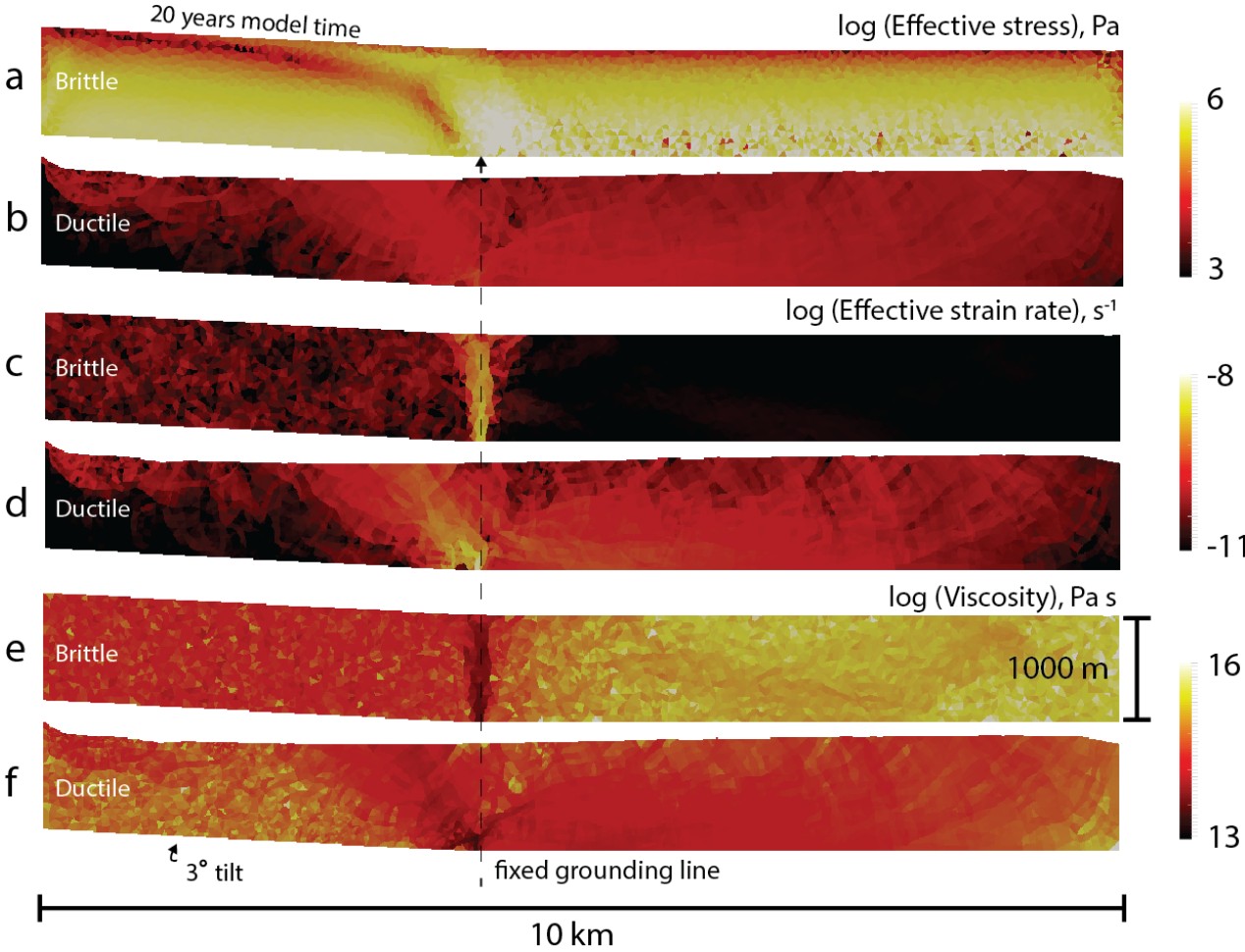

5    **Figure 3:Comparisons of purely brittle (a, c, e) and purely ductile (b, d, f) tilted slab experiments. (a, b) Effective stress, (c, d) effective strain rate, (e, f) viscosity after 20 years model time. Overall the brittle rheology results in a narrow process zone of high stress and deformation, while the ductile rheology has a more diffuse area of strain and lower stresses.**

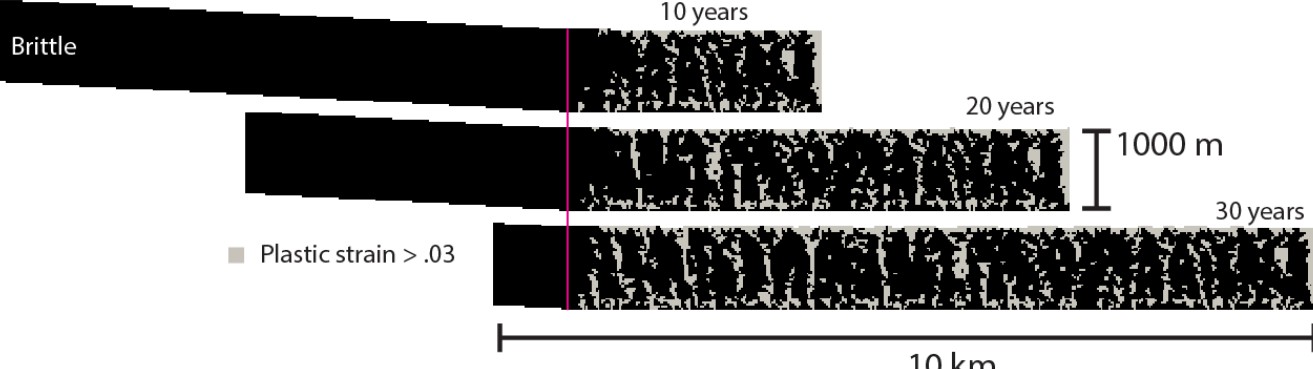

**Figure 4:** Simulation of brittle ice slab being advected down the inclined plane throughout the model time. Pink vertical line denotes the change in the angle of applied velocity boundary conditions from 3 degrees to flat. Black ice indicates completely intact ice, while grey represents ice that has failed. Grey vertical lines appear with regularity.

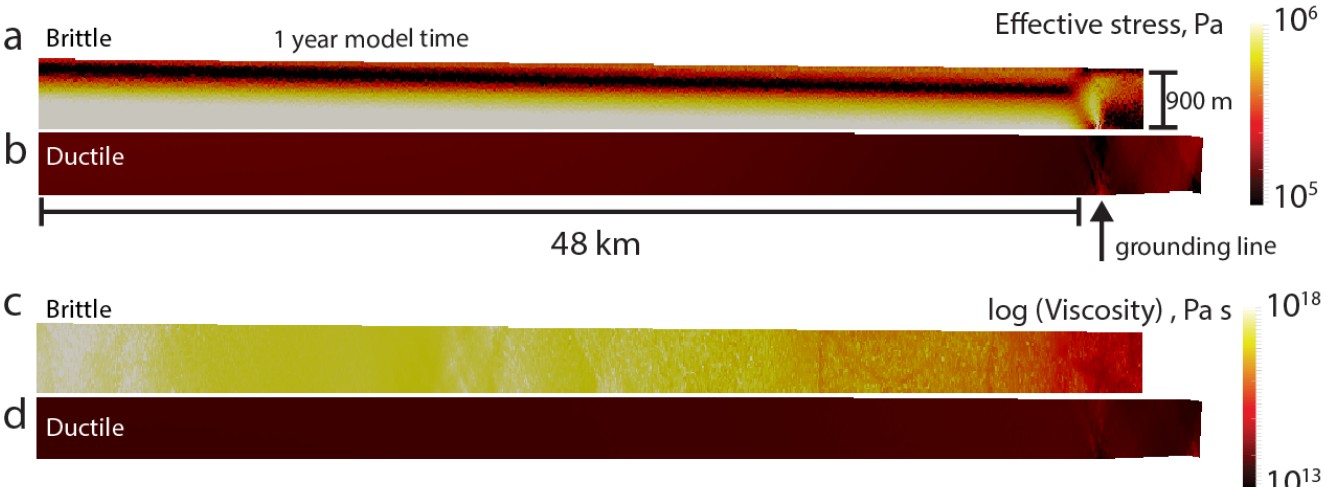

**Figure 5:** Effective stresses in (a) purely brittle and (b) purely ductile ice for a wedge experiments after 1 year of model time. The left nodes vary freely in the vertical and are fixed to 0 in the horizontal, and the basal boundary is free-slip. There is an order of magnitude difference in stress. Again, brittle ice experiences a small process zone of high stress immediately at the grounding line (black arrow) where there is a transition from freely slipping ice to a buoyancy stress condition with a 1 m diurnal tidal forcing, and the forces due to bending are high. Ductile ice also experiences an increase in stress at this location, although the transition is not as sharp. Most notably, flow in brittle ice is negligible compared to ductile ice after 1 year, owing to the very large (as much as 5 orders of magnitude) difference in viscosity between the (c) brittle and (d) ductile ice.

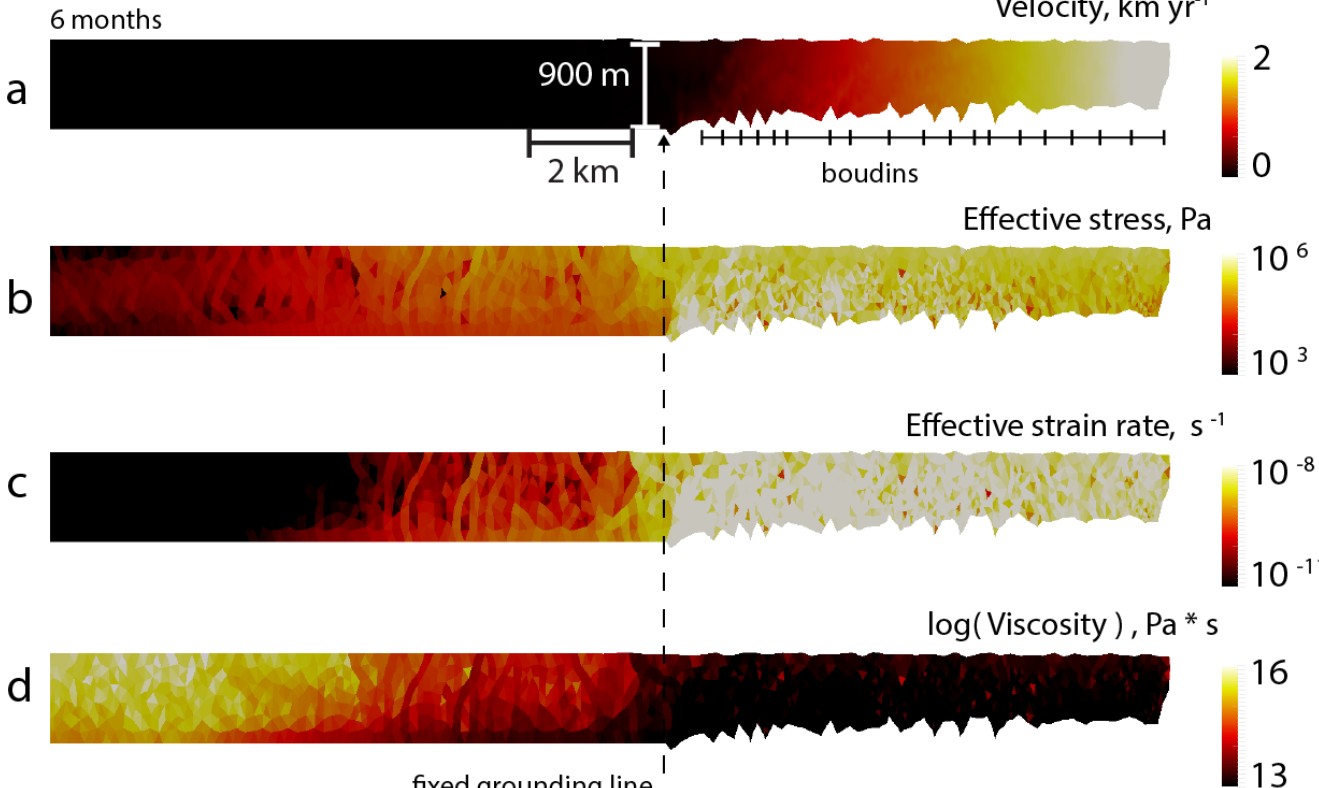

**Figure 6:** Simulation of semi-brittle wedge of ice after 6 months model time, undergoing a jump in boundary conditions from freely slipping on the basal side (left of grounding line – dashed arrow) to floating in the ocean (right of grounding line). The horizontal velocity (a) is fixed at x = 0 and free elsewhere, while the vertical velocity is free at x = 0 and fixed to zero up to the grounding line. The gradient in thickness drives the flow of ice over the grounding line where ice experiences a transition to stress boundary conditions that represent floatation and 1 m tides. Dashed bars in the floating tongue show the development of boudins. The effective stress (b) and strain rate (c) fields both show orders of magnitude increases at the grounding line: the jump in strain rate allows the ice to be evaluated as brittle in DES and the associated stresses are high enough to reach yield (see Fig. 6). The corresponding viscosities (d) just upstream of the grounding line are high and decrease 3 orders of magnitude as the ice expands out into the ocean under its own weight.

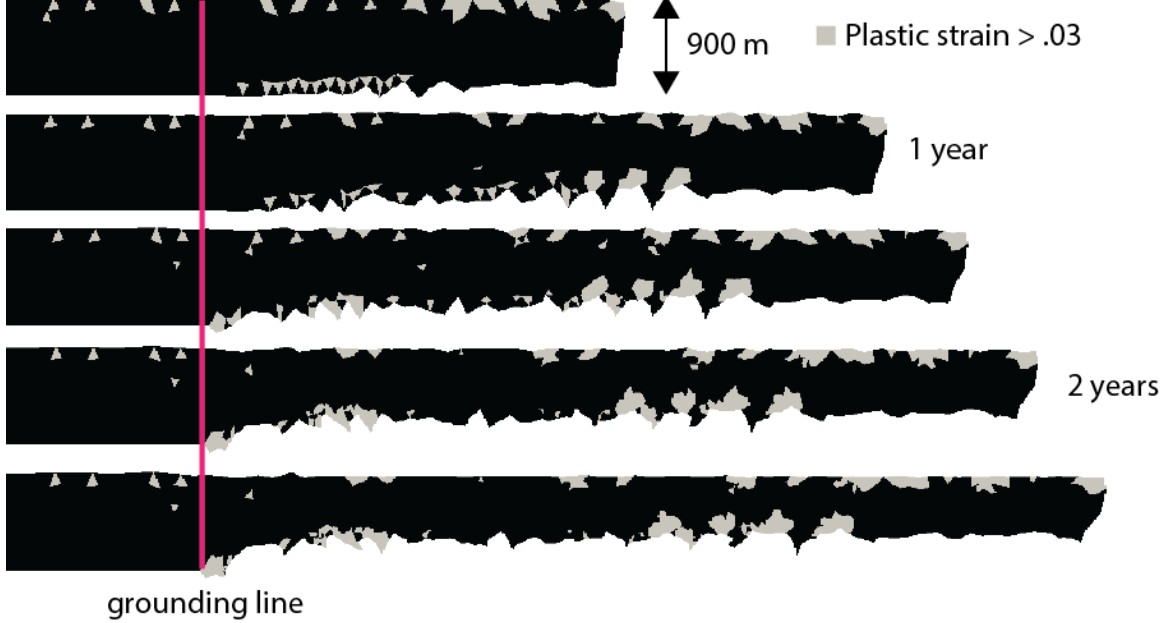

**Figure 7: Plastic strain over 2.5 years model time. Ice fails in tension at the surface near the terminus with regularity at the grounding line where hydrostatic stress is applied (in pink). Ice forms boudin-like features after accommodating a large amount of strain.**

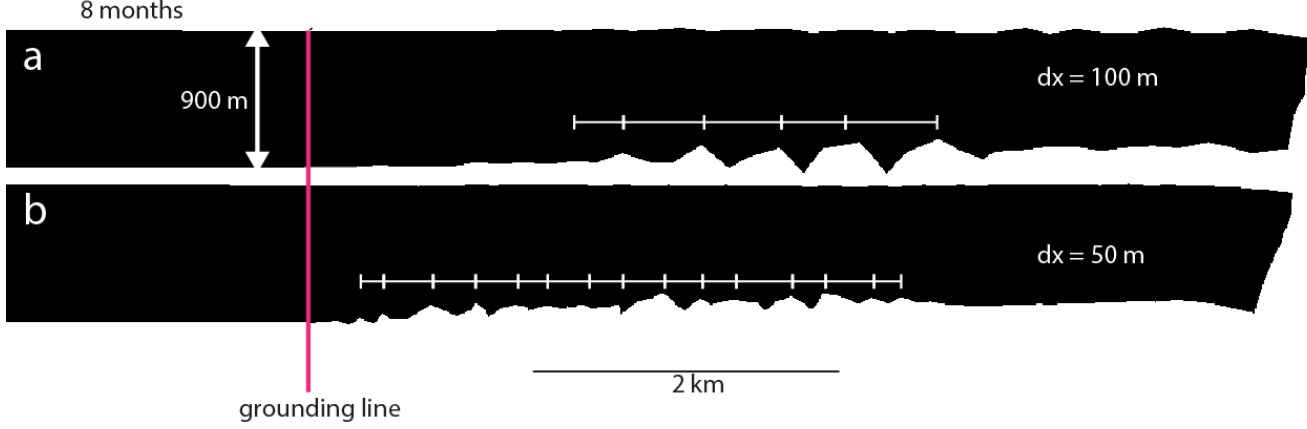

**Figure 8: Geometries of the mesh for a resolution of (a) 100 m and (b) 50 m. The pinch and well features ("boudins") appear with regularity (white bars) and decreasing size upon mesh refinement. Computational limitations precluded running the simulations at quartered resolution.**

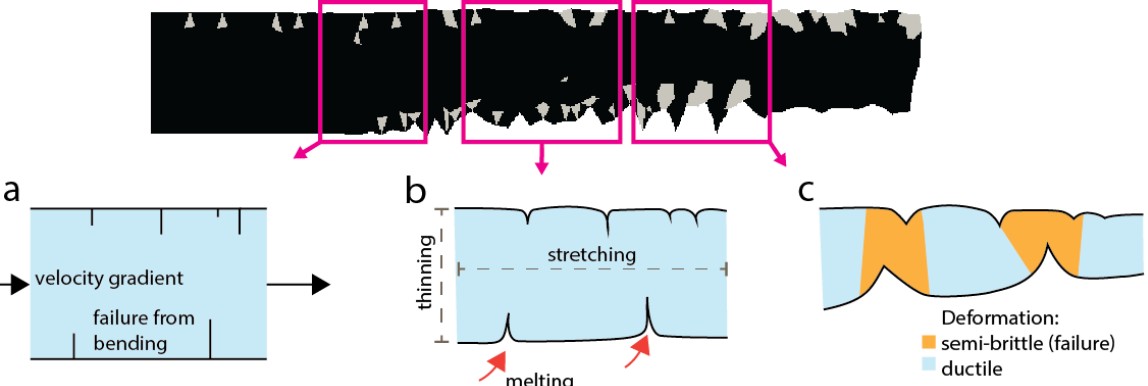

**Figure 9: Schematic of how semi-brittle deformation could proceed in nature, through space and time. (a) Initial failure forms due to high strain rates as ice accelerates across the grounding line, with the added bending due to tidal motion. (b) Ice continues to accelerate as it floats without resistance into the ocean; (not simulated here) melting from hot, buoyant water enters cracks and erodes crack walls, widening and thinning the ice. (c) Ice reaching its terminal speed at the front undergoes both semi-brittle deformation and ductile deformation. Semi-brittle failure occurs where ice has previously failed (in thin spots) and further thins the floating tongue, while ice between surface cracks and bottom cracks undergoes ductile deformation. The geometry produced by these processes resembles boudins, which eventually calve into the ocean.**

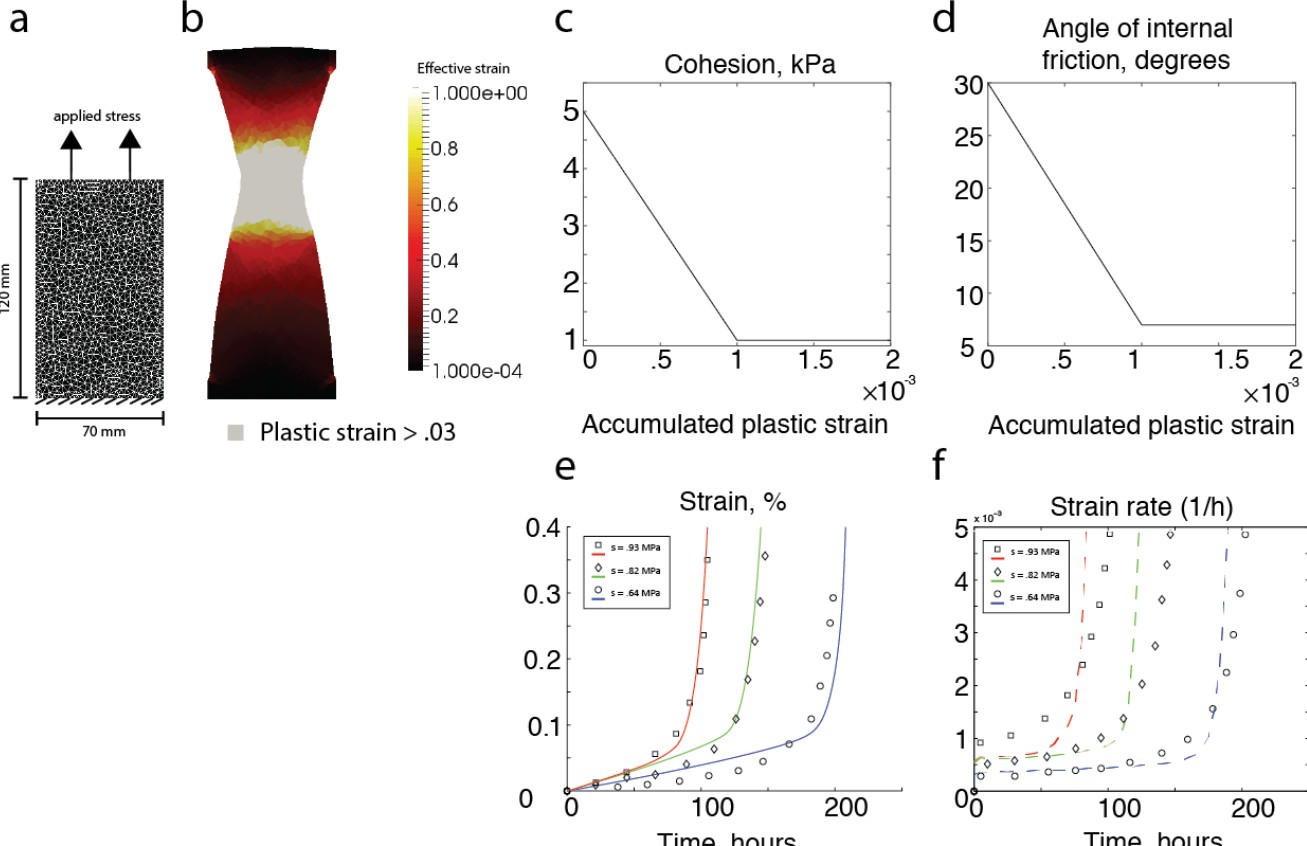

**Figure 10: Calibration experiments for semi-brittle ice based on laboratory-derived data presented in Mahrenholtz and Wu (1992). (a) Initial mesh and dimensions and (b) effective strain after failure of semi-brittle ice with 0.82 MPa applied vertical stress. Overlaid on top is the accumulated plastic strain (grey) of elements > 0.03. Because the grey extends horizontally throughout the ice plug we consider an accumulated plastic strain of 0.03 or greater to represent ruptured ice. (c) Cohesion and (d) angle of internal friction parameters that are needed as a function of accumulated plastic strain to reproduce the (e) strain and (f) strain-rate-versus-time curves presented in Mahrenholtz and Wu (1992). Open black points are laboratory-derived data and coloured lines are from DES experiments.**

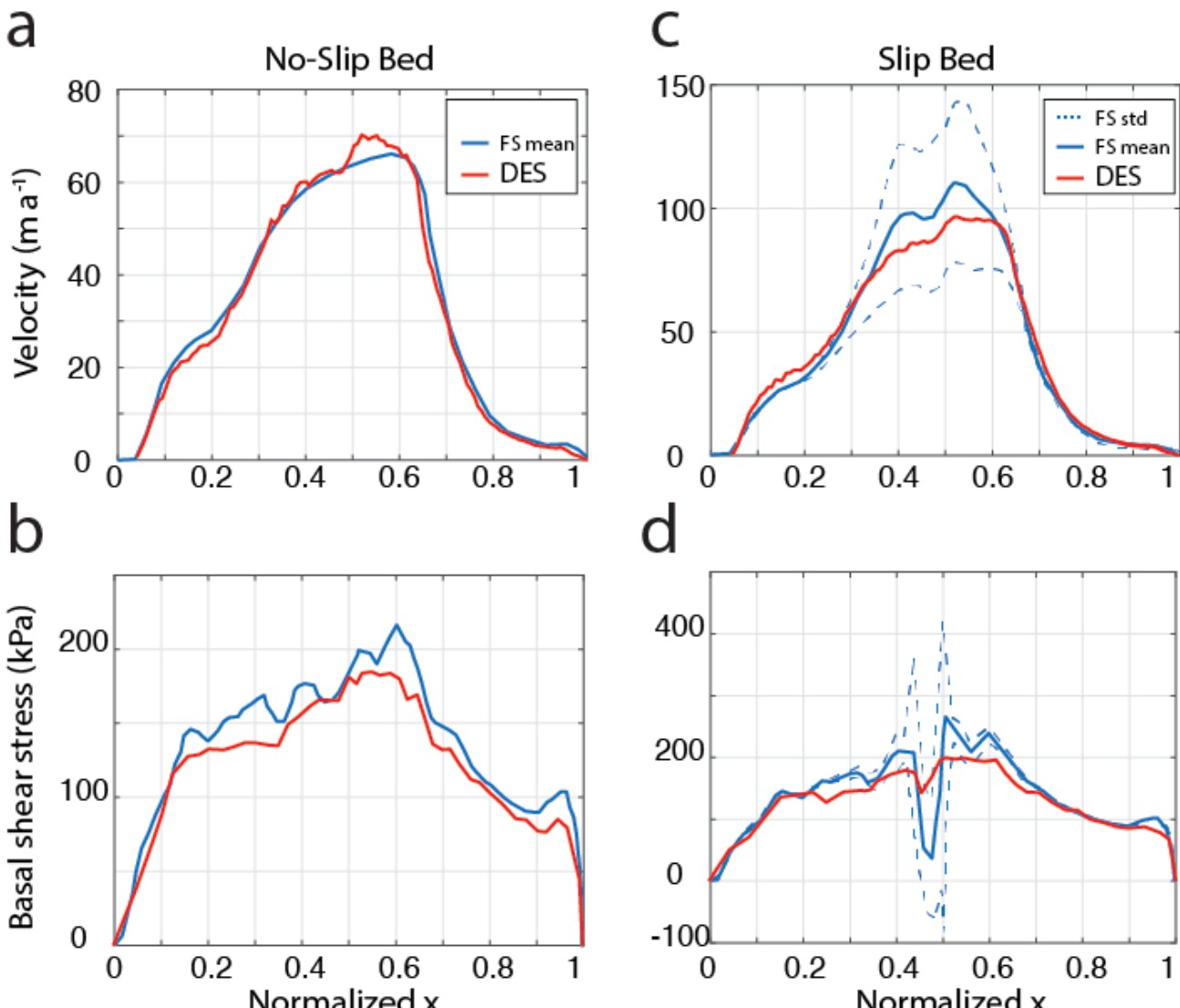

**Figure 11: DES with semi-brittle ice executed according to Experiment E presented by Pattyn et al. (2008). (a) Surface velocity and (b) basal shear stress for ice completely frozen to the bed, and (c) surface velocity and (d) basal shear stress for ice with a freely slipping patch for a small portion of the domain. Full stokes (FS, blue line) model means compare reasonably with those from DES (red line).**

