# Peer review of "Semi-brittle rheology and ice dynamics in DynEarthSol3D"

_The Cryosphere, 2016_

## Referee Comment (RC1) · Anonymous Referee #1 · 27 Jun 2016

**The Cryosphere** - **TC2016-88** *"Semi-brittle rheology and ice dynamics in DynEarthSol3D"* by Logan and others.

This paper present a new modeling framework to account for the semi-brittle (brittle and ductile together) behaviour of ice using the DynEarthSol3D model. Two applications are proposed, the first is a comparison of the brittle and ductile behaviours and the second an experiment with the semi-brittle behaviour. These two experiments are performed using very simplified setup (geometry and boundary conditions). I have three main concerns with this paper. I have also listed some minor concerns/typos that should be accounted for in a revised version.

**Boundary conditions** I am wondering how much the conclusions from the first experiment are related to the imposed kinematic boundary condition, especially the two orders of magnitude difference between brittle and ductile effective stress. Imposing a velocity field on three of the four boundaries of the domain conduct to stress that are not realistic at all. The flow of ice is gravity driven in reality so that I am not sure of what can be really learned from this first experiment. In other words, I am not sure that under realistic conditions (realistic geometry and boundary conditions) the two approaches would give so different stress field (because the global static equilibrium would be similar if not the same).

**Description of the implemented rheology.** The paper should be really improved regarding the presentation of the implemented rheology and failure criteria in the model. All this material should be consistently presented in section 2.1. Some of these aspects are described all along the manuscript whereas they should really be presented consistently in the model description section (e.g., the Mohr-Coulomb failure envelope, the fact that there is no failure criteria for the ductile behaviour or the expression for the ice effective viscosity are presented in the application section). Some part of the model are not described at all. For example, it is not clear what becomes the rheology when the Morh-Coulomb criteria is reached in the brittle approach? From Fig. 4, I understand that in fact there is no real failure of the material property of ice for the brittle rheology and that failure is estimated when a plastic strain is larger than 0.03? This should be really explained in much more detail.

**Sensitivity to mesh quality** The free surface for both experiments looks very jagged. It is mentioned page 9 line 16 that it is an artifact of the low resolution and that these features disappear with higher resolution. I don't understand then why the results with a higher resolution are not presented, especially when it is mentioned in the conclusion (page 13, line 23) that all the simulations presented here are computationally cheap! How much the results presented in this paper are mesh dependent? The geometries obtained after only 6 months of simulation and presented in Fig. 5 look so bad that I have some doubts that the simulation can be performed for a longer time before exploding? It looks like you have positive slope which would induce reverse velocity for a 2D flow line problem. How much the spacing of the "crevasses" presented in Fig. 4 is mesh dependent? All these feature (distribution of the plastic strain larger than 0.03, upper and lower surface ondulations) seem to be of the element size. Information regarding the mesh are really needed, as well as a clear study of the sensitivity of the results to mesh refinement.

**Other remarks**

page 2, line 8: add "e.g." in front of these references as they are not exhaustive on that subject. The same remarks apply at other places in the manuscript.

page 2, lines 10-14: the tone of this introduction is a bit naive? You are writing in TC, people have heard about calving?

page 2, line 17: there is nothing about LEF mechanics in Larour et al. (2012) paper.

page 2, line 19: or a mixture of both like in Krug et al. (2014).

page, line 23: over *very* short time scales?

Equation (1): define $\sigma_e$ as well. The definition of the effective pressure should be presented here.

page 3, line 11: Most *ice-flow* numerical models

page 3, line 18: I don't agree that viscous model are not capable to represent ice failure and ice retreat. As far as I know (and some of these paper are cited in the present manuscript), there have been some work to include these processes.

page 3, line 26: the need of elastic stress to be accounted for is a bit affirmative and, as it is said in this paper, would need some modeling effort to really understand how important it is to account for them. Moreover, I think it really depends at which scale (time and space) you are interested, which should be mentioned.

page 4, line 14: some words in the introduction about particle models or discrete element models would be interesting (e.g. Bassis and Jacobs, 2013; Åström and others, 2014), and how they compare to the present approach.

page 4, line 17: the main issue of using a Lagrangian approach in glaciology relies in accounting for the in/outcoming flux of ice on the domain boundaries (accumulation and/or ablation on the surface, melting/accretion at the base). You should mention in the manuscript how this problem is (or will) be overcome for realistic applications.

page 4, line 19: FS models neglect acceleration because it is completely negligible for the time step of interest of many applications. In the proposed applications, it would be interesting to document the relative contribution of acceleration in the total momentum. Their importance, as stated here, has still to be proven?

page 5, line 5: avoid repeating "*of ice*".

page 5, line 7: from my experience, a Dirichlet BC is only required where you have an output flow and not on all the boundaries, as it seems the case here. Does it come from the Lagrangian formulation?

page 5, line 13: This sentence is not clear and looks technical more than related to the physics in the model? Which equation is solved for incompressibility should be given here, whereas how it is solved should be given in the following section.

page 5, line 23: it is not clear if the floatation is fulfilled for the floating part?

page 5, line 25: you mean an explicit time-stepping scheme?

page 6, line 6: are given in Choi et al. [2013].

page 6, line 14: no need to define again the minimum element facet length.

page 7, line 25: I don't really think this list of capabilities is relevant for the present paper

page 8, line 4: we divide this section (delete *the*)

page 9, line 4: the two order of magnitude differences in term of stress certainly is the result of the very particular boundary conditions applied here and therefore no real conclusion can be drawn from this setup regarding a realistic case (see major remarks).

page 9, line 5: (Figs. 3a and b)

page 9, line 9: (Figs. 3e and f)

page 9, line 16: So why not showing these better results obtained with an higher resolution? In any case, a sensitivity study of our results to the mesh resolution would clearly improve the strengh of the paper.

page 9, line 25: How much the spacing shown in Fig. 4 is dependent of the mesh. In other words, do you get the same spacing with a mesh with halved elements?

page 10, line 23: The most appropriate variable to write a criteria for damage would be the Cauchy stress, not the strain or strain-rate.

page 12, line 9: How would you account for basal melting in a Lagrangian model?

The basal geometry from Fig. 3 does not correspond to the setup presented in Fig. 2a. In Fig 2a it is a straight line over 10 km whereas in Fig 3 there is two lines that define the base (over the same 10 km)? I am wondering if the 10 km scale indicated in Fig. 3 is therefore correct? An horizontal scale in Fig. 4 would be helpful for the same reason.

The geometry in Figs. 5 and 6 look very mesh dependent and it would require some convincing arguments (i.e., a mesh sensitivity study) before moving to physical explanations about these modeled features as done in Fig. 7.

**References**

Åström, J. A., D. Vallot, M. Schäfer, E. Z. Welty, S. O'Neel, T. Bartholomaus, Y. Liu, T. Riikilä, T. Zwinger, J. Timonen and others. 2014, Termini of calving glaciers as self-organized critical systems. *Nature Geoscience*, **7**(12), 874–878.

Bassis, J. and S. Jacobs. 2013, Diverse calving patterns linked to glacier geometry. *Nature Geoscience*, **6**(10), 833–836.

---

## Referee Comment (RC2) · J. Bassis (Referee) · 18 Jul 2016

General appreciation: This study seeks to apply a viscoelastic rheology combined with a semi-brittle yield strength to simulate the flow and failure of glacier ice. The authors apply a pre-existing model designed to simulate deformation of the solid Earth and apply it to simulate failure that originates near the grounding line of glaciers that transition from grounding to floating. The authors perform experiments using idealized geometries to demonstrate the method is promising and predicts basal crevasses that originate downstream from the grounding line.

Overall, I think that the approach proposed is innovative and has potential to provide significant insight into calving and fracture processes of glaciers and ice shelves. I do, however, have some comments and lingering questions. Some of the comments relate

to pleas for additional clarification of aspects of the model. Others are suggestions for additional model experiments. As is always the case, reviewers want a different subset of experiments than those the authors provide. Given limited computational resources, I don't think it is necessary the authors perform all of the additional simulations suggested, but the authors should consider a subset of these experiments, if not for this study than for future studies. Many of the comments are geared towards clarifying concepts and approaches to make the manuscript more accessible for a wider readership of glaciologists and the authors should at least consider some of the modifications. These suggestions are described in more detail below

1. Rheology and yield relations. I would like to see a much more detailed description of the spectrum of rheologies and yield relations used. The authors provide a description of the usual power-law viscous creep deformation glaciologists are used to, but few equations describing the rheology beyond this. I recognize that the model used is fully documented in prior publications. However, the authors are introducing concepts that are new (or at least less familiar) to glaciologists and some hand holding is appropriate. There are also some details that are missing. For example, the 2D viscoelastic simulations are presumably done under plane stress or plane strain conditions, but I could not find which in the manuscript. (I apologize to the authors if I missed this in the manuscript.) More importantly, I would like to see equations describing the yield relations and some description of the assumptions. For example, the authors state that they use a Mohr-Coulomb yield strength. The typical interpretation of the Mohr-Coulomb yield strength is that materials fail when the maximum shear stress exceeds a threshold that depends on the normal stress and a cohesion parameter. This is occasionally interpreted as the initiation of new faults or the re-activiation of previously existing faults. Which interpretation are the authors assuming? Or does it not matter? Also, what happens above the yield strength? Does the yield strength denote a boundary between flow laws, as in a Bingham plastic? What happens once ice has failed? Does it return to behaving like intact ice if the stress decreases beneath the yield strength (as is true in a granular material) or does it continue to behave as damaged ice once yielded, irrespective of the current state of stress?

Another question I have relates to tensile versus shear failure. For example, typically, we think of crevasses as tensile failure features, but the Mohr Coulomb failure envelope is usually applied to shear failure. (In the absence of a cohesive strength, a Mohr-Coulomb failure law implies no tensile strength.) How do the authors simulate tensile failure as opposed to shear failure? Are there different yield strengths used? Typically, faulting is more important in the Earth, but in ice people often focus on tensile failure. (We partially dispute this. See for example, Bassis and Walker, 2012, Proceedings of the Royal Society.). Moreover, failure envelopes in compression and tension are usually very different with compressive strengths much larger than tensile strengths. Is this accommodated in the model? Is compressive failure considered negligible?

There are also some technical questions associated with simulating yielded ice. We (and others) have found that the maximum shear stress criterion associated with Coulomb-like failure can be difficult to implement numerically. Instead, we (and many others) often prefer to use the effective stress (2nd deviatoric stress invariant). This is qualitatively similar, but corresponds to a Drucker-Prager granular material and not a Coulomb-granular material. I assume the authors are using the Coulomb criterion, but do the authors need to stabilize the method to avoid the numerical errors associated with the non-robustness of finding a maximum?

All of these questions leave me with an imperfect understanding of the physics assumed by the authors and this clouds my understanding of the results that follow from these assumptions. I suspect most readers will have similar questions and it will help tremendously if the authors step us through the assumptions and assumed physics instead of rushing us through to the results. In many ways, I think the physical model has much greater value than the preliminary results so I urge the authors to take the time to explain the model thoroughly to the audience.

2. Boundary conditions. The authors specify velocity boundary conditions at the left,

bottom and right edges of the domain. Specifying a velocity boundary condition at the bottom is a bit odd. Typically, we would specify a sliding law or, alternatively no-slip or free-slip boundary conditions. I'm a little bit worried that the velocity boundary condition contaminates the results. I would recommend either re-running simulations using a sliding law. We often like to do both the free-slip and no-slip conditions to bracket behavior when doing idealized experiments where we don't want to specify parameters in a sliding law. If this is unfeasible, then I think some additional justification for the boundary conditions is appropriate. If the authors maintain the velocity boundary condition the authors should plot basal shear stress. Basal velocities are specified to be reasonable, but does this produce realistic basal shear stresses? The authors also might want to consider using a free-slip boundary condition for the vertical displacement in the left side of the domain. This will avoid the weird abrupt decrease in ice thickness near the left wall.

3. Model numerics and comparison with existing solutions. The model that the authors are using is a complex viscoelastic model used to study solid Earth deformation. The model appears to have been well benchmarked agains standard solutions and so hopefully the model numerics is well understood. However, there are aspects of the numerics associated with the flow of ice that are not as well represented in the previous set of benchmark experiments. In particular, the mass weighting and damping to obtain stable solutions in the explicit integration of the Navier-Stokes equations (with inertia) does not appear to have been calibrated with ice is mind. This raises some questions about the appropriateness of the numerical parameters. The mass weighting method that the authors use to time step the Navier-Stokes equations is one of the those methods that gets periodically rediscovered. I would personally prefer if the authors made it clear that the mass weighted explicit integration is used as a means of avoiding the cumbersome and expensive task of solving of large-non-linear sets of equations and that individual time steps do not provide accurate solution to the equations of motions. The hope is that over long time scales the solution is approximately steady-state, which corresponds to the Stokes equations that the authors rely want to solve. Presumably,

one could use, say, a multi grid or other fancy numerical solver instead to find the solution to the elliptical set of equations. Having said this, it would be nice if the authors could show that the model that they use is able to reproduce existing analytic or benchmark solutions for glacier flow. There have been a number of model inter comparisons that the authors could consider. I'm agnostic to the choice, but it would be reassuring to show that under viscous conditions, the authors can reproduce standard solutions for velocities and ice thickness. The authors have probably already done this and so a few sentences or a section in an Appendix would be all that is required. If possible, it would be great to see some convergence studies to show that the results shown in the paper are not numerical artifacts or signs of instabilities. The figures in the paper show jagged ice shelves. I suspect that failure will look more realistic if the authors conduct higher resolution model runs.

4. Interpretation of model results: One of the most intriguing results that the authors obtain is that they produce basal crevasses under ice shelves. We tried to explain these features in a recent paper using a perturbation approach (Bassis and Yue, 2015, EPSL). We focused on viscous instead of brittle ice and found a long wavelength instability that could result in wide basal crevasses so long as the stress was sufficiently large compared to the confining pressure. In our formalism, we can also examine brittle failure by taking the limit that the flow law exponent (n) tends to infinity. When we do this we find that the dominant wavelength is of the order of the ice thickness. The growth rate of perturbations, however, becomes extremely large. This is a consequence of the fact that in our model, we assume the ice is isothermal. This implies that over long wavelengths, the strain rate and deviator stress are both constant with depth and the entire ice shelf reaches the yield strength at the same time. This raises the question of whether the results here are consistent or inconsistent with our (admittedly limited) analytic result? If not, what controls the rate at which brittle failure propagates. What control the spacing between basal crevasses? Incidentally, the perturbation analysis that we conduct is analogous to some of the original perturbation calculations to explain boudinage in rock by Smith and others.

5. Clarification of the role of elastic stresses: The authors make a really interesting point that despite the fact that elastic stress decay over long time scales, the fractures that result from elastic stresses remain important. Based on this, the authors argue that we need viscoelastic rheologies to accomodate failure. I don't disagree with the authors. However, if elastic stresses are important (through their role in promoting failure) then, unlike purely viscous flow, simulations become an initial value problem. What I mean by this is that in purely viscous flow we can initialize a model with an unrealistic initial condition. The unrealistic initial condition will generates shocks in the model that will relax over time and we typically either initialize a model in such a way as to not generate shocks or allow the model to spin up until those shocks have sufficiently dissipated that the model is no longer contaminated by these shocks. In a viscoelastic model with failure, it seems possible that the template for failure will be strongly controlled by the initial condition—especially if the initial condition is unrealistic and generates shocks. The authors are starting with simple wedges and allowing them to evolve. Do the authors obtain similar results if the model is first spun up to a quasi-steady state consistent with purely viscous flow and only then is failure allowed to occur? Do elastic stresses remain important if the model is started from a configuration in which elastic stresses have already decayed? What is an appropriate starting condition for models or is the initial condition not that important?

Incidental comments:

Page 3, near line 5 "Ductile fracture is initiated by the formation of distributed voids that eventually coalesce to form a macroscopic fracture". Laboratory experiments indicate that ductile failure growth through the nucleation and growth of voids does not occur in ice. Fractures instead usually propagate through the formation and propagation of micro-cracks. I think that is what we proposed occurs ahead of the rift in the Amery Ice Shelf. The void growth mode of failure occurs in metals (perhaps rocks as well?), but to my knowledge is inapplicable to ice under terrestrial conditions. There is, of course, the separate question of whether the macroscopic behavior of ice in glaciers can be

simulated using a framework appropriate for ductile failure of metals. However, I would like the distinction to be made more clearly in the manuscript.

Page 8, left, right and bottom velocities are set to 300 m/a. First, I recommend using more physical notation, like inflow, outflow and basal boundary conditions, including left, right, bottom as the authors see fit. Second, the fact that the velocity is constant implies no bulk extensional stresses, which seems odd for a glacier. I would appreciate more description for the motivation for this set of experiments.

I'm less confident for the evidence of a sharp brittle-ductile transition at a critical strain rate. We clearly see tensile fractures at a range of strain rates, with the controlling variable usually stress. Of course, stress and strain rate are interchangeable if the ice is isothermal, but that is not often the case.

Page 1 Line 15: "We find that the use of a semi-brittle constitutive law is a necessary material condition to form the . . ." I believe necessary should be replaced with sufficient. I don't think the authors have proven that no other conditions are able to reproduce fields of basal crevasses. What they have demonstrated is that a brittle rheology is sufficient to produce this feature.

Page 3 Line 5: Usually brittle failure of ice is thought to be a consequence of high stresses rather than strain rates. See, e.g., Vaughan, Journal of Glaciology, 1993 "Relating the occurrence of crevasses to strain rates".

Page 3 Line 5: The point about ductile failure versus brittle failure is subtle. The coalescence of voids to form macroscopic fractures might actually be brittle. At the very least, the formation of these voids appears to be seismic. But the brittle failure that occurs may act like plastic or ductile failure over macroscopic length scales.

Page 3, Line 20 It seems odd to claim that models based on Linear Elastic Fracture Mechanics do not predict the correct stresses if their rheology is assumed to be purely viscous. By definition the "E" in LEFM corresponds to elastic so how can the rheology

be assumed to be purely viscous?

---

## Author Comment (AC1) · 12 Sep 2016

**"Semi-brittle rheology and ice dynamics in DynEarthSol3D"**

L. C. Logan, L. L. Lavier, E. Choi, E. Tan, G. A. Catania

The authors truly want to express gratitude to the reviewers, appreciating the time it takes to put forth the thoughtful and incisive comments presented within. To summarize: we have largely engaged with all the reviewers' questions and requests, and hope that our responses satisfy. This manuscript revision includes 2 appendices meant to respond to and allay concerns expressed by the reviewers, as well as some additional model results presented when computational resources allowed. We sincerely believe that the reviewers' comments greatly strengthened this manuscript. Many thanks, and best wishes, Authors

Format for reading response: *Referee comment* (**R1** or **R2**) Author response (**AR**) Change to manuscript (if appropriate)

**Author response to reviewer 1:**

**R1: Boundary conditions** I am wondering how much the conclusions from the first experiment are related to the imposed kinematic boundary condition, especially the two orders of magnitude difference between brittle and ductile effective stress. Imposing a velocity field on three of the four boundaries of the domain conduct to stress that are not realistic at all. The flow of ice is gravity driven in reality so that I am not sure of what can be really learned from this first experiment. In other words, I am not sure that under realistic conditions (realistic geometry and boundary conditions) the two approaches would give so different stress field (because the global static equilibrium would be similar if not the same).

- **AR:** You bring up valid concerns that are also expressed by Jeremy Bassis (reviewer 2), and we agree that the glacial implications from this first experiment are limited and extrapolations from these experimental results must remain tempered. We clarify the purpose of Experiment 1 in the text: to show the difference in stresses but also to see the evolution of failure in translating ice. We clarify that we do not attempt to extrapolate realistic values for basal crevasse spacing from the brittle experiment, and say explicitly that this experiment is demonstrated only to depict a qualitative behavior, which motivates the second set of experiments. We also include the figure below in the manuscript, which should ease the worries of the concerned reader regarding the magnitude of stresses in the ductile vs. brittle rheologies.
  - 1. See new Section 2.2
  - 2. See new Figure 5

However, it is important to understand that the flow in the experiments is very much driven by gravity. The pressure gradient due to the incline drives flow at ~470 Pa/m (Wu and Lavier, 2016, eq. 1). This is sufficient to generate flow in a 1000 m thick ice layer of viscosity ~1013 Pa s. The resulting strain rate *due to this pressure gradient alone* ranges from  $10^{-9}$  to  $10^{-7}$  s-1, enough to decrease the viscosity to  $10^{12}$  Pa s, assuming Glen's flow law. In addition, the constant velocity at the bottom of the box and on the sides does not generate any shear strain rate (being of constant velocity throughout the entire model domain). Therefore, these boundary conditions form a *pseudo-rigid* box, and do not generate additional flow.

Put differently and succinctly, the box translates as a rigid entity in which the fluid flow is driven by gravity. The only location where the boundary conditions explicitly (and purposefully) impact deformation is at the bending fulcrum ("grounding line"). There the strain rate changes rapidly in the materials to accommodate the change in shape of the box. Down flow of the fulcrum, the pressure gradient is negligible: correspondingly, the internal flow due to gravity decreases as does the strain rate. The resulting viscosity then increases to >  $10^{14}$  Pa s. The ice essentially becomes rigid for both rheologies (ductile and brittle) down flow of the bending fulcrum. Thus the ice in the inclined portion of the domain is gravity driven and the stresses are realistic. Additionally, brittle (elastoplastic) ice only deforms due to bending stresses imposed at the fulcrum, which are fundamentally much higher than viscous stresses.

In addition, to further allay concerns that the velocity boundary condition does not contaminate the results, we show results from Experiment 2 (the flat, frictionless wedge) with purely brittle and ductile rheologies. Here is that image after 1 year of model time (this is now included in the manuscript):

**New Figure 5:** Experiment 2, effective stress in [a] brittle ice and [b] ductile ice after 1 year model time. The bottom of the ice is frictionless, and there is no inflow ice velocity. Ice flows [a] (or does not, [b]) from left to right. There is an order of magnitude stress difference between the two rheologies and the ductile ice flow is driven entirely by thickness. Brittle ice *would flow* to the right, but the thickness gradient is not large enough, and the corresponding viscosity is sufficiently high to make flow velocities much smaller than those of ductile ice. *Imposed velocity conditions* and bending *are*

*needed* (as in Experiment 1 – tilted planar) to observe regularly spaced zones of vertical failure. Pink arrow is transition to buoyancy stress boundary condition.

Again, flow here is driven *entirely* by gravity: the thickness gradient causes a pressure gradient which drives flow to the right, and the downstream portion (right side) of the domain undergoes floatation stresses at the black arrow. The order of magnitude stress difference is readily apparent, and in no way can be attributed to contaminating boundary conditions. After 1 year of simulation the brittle ice (Fig. 1a) has not advanced over the grounding line (although it has failed there due to bending buoyancy). That the brittle ice does not flow is the reason we executed the first set of experiments: not only to show the difference in stresses, but also to show the temporal evolution of failure of ice as it goes through a bend. In the image above, the brittle ice is simply too strong (and the pressure gradient too low) to drive flow across the grounding line.

**R1: Description of the implemented rheology**. The paper should be really improved regarding the presentation of the implemented rheology and failure criteria in the model. All this material should be consistently presented in section 2.1. Some of these aspects are described all along the manuscript whereas they should really be presented consistently in the model description section (e.g., the Mohr-Coulomb failure envelope, the fact that there is no failure criteria for the ductile behaviour or the expression for the ice effective viscosity are presented in the application section). Some part of the model are not described at all. For example, it is not clear what becomes the rheology when the Morh-Coulomb criteria is reached in the brittle approach? From Fig. 4, I understand that in fact there is no real failure of the material property of ice for the brittle rheology and that failure is estimated when a plastic strain is larger than 0.03? This should be really explained in much more detail.

- **AR:** This is an easy fix: we kept most of the details out at first because they were presented in Choi et al. (2013), when the model was first published. But we are happy to include these details again and agree that their presentation strengthens the manuscript. Additionally, so that readers can be assured that the values used in material properties are appropriate we included an Appendix A on the calibration experiments we used to validate the semi-brittle material.
  - 1. See new Section 2.2
  - 2. See Appendix A (semi-brittle ice calibration experiments)

**R1:** Sensitivity to mesh quality The free surface for both experiments looks very jagged. It is mentioned page 9 line 16 that it is an artifact of the low resolution and that these features disappear with higher resolution. I don't understand then why the results with a higher resolution are not presented, especially when it is mentioned in the conclusion (page 13, line 23) that all the simulations presented here are computationally cheap. How much the results presented in this paper are mesh dependent? The

geometries obtained after only 6 months of simulation and presented in Fig. 5 look so bad that I have some doubts that the simulation can be performed for a longer time before exploding? It looks like you have positive slope which would induce reverse velocity for a 2D flow line problem. How much the spacing of the "crevasses" presented in Fig. 4 is mesh dependent? All these feature (distribution of the plastic strain larger than 0.03, upper and lower surface undulations) seem to be of the element size. Information regarding the mesh are really needed, as well as a clear study of the sensitivity of the results to mesh refinement.

- AR: Failure in ice is marked by localized strain, and strain localization is well-known to be mesh-dependent under rate-independent plasticity (the brittle rheology in DynEarthSol3D). Usually what people mean by mesh-dependence in this context is that the width of a band of localized strain is determined by element size and/or the orientation of the band tends to follow "grains" of a mesh even though they are not consistent with stress field. We present experiments with halved resolution. Computational resources did not allow for a presentation of quartered resolution.
  - 1. See new Figure 8

**Other remarks**

**R1:** page 2, line 8: add "e.g." in front of these references as they are not exhaustive on that subject. The same remarks apply at other places in the manuscript.

AR: Ok.

**R1:** page 2, lines 10-14: the tone of this introduction is a bit naive? You are writing in TC, people have heard about calving?

AR: Ok.

**R1:** page 2, line 17: there is nothing about LEF mechanics in Larour et al. (2012) paper.

**AR:** You're right: it's a 2004 paper. Cited.

**R1:** page 2, line 19: or a mixture of both like in Krug et al. (2014).

**AR:** Krug et al. (2014) is cited later. We moved it up though.

**R1:** *page, line* 23: *over very short time scales?*

AR: Ok.

**R1:** Equation (1): define  $\sigma e$  as well. The definition of the effective pressure should be presented here. **AR:** Ok.

**R1:** page 3, line 11: Most ice-flow numerical models

AR: Ok.

**R1:** page 3, line 18: I don't agree that viscous model are not capable to represent ice failure and ice retreat. As far as I know (and some of these paper are cited in the present manuscript), there have been some work to include these processes.

AR: Rephrased.

**R1:** page 3, line 26: the need of elastic stress to be accounted for is a bit affirmative and, as it is said in this paper, would need some modeling effort to really understand how important it is to account for

them. Moreover, I think it really depends at which scale (time and space) you are interested, which should be mentioned.

**AR:** The papers suggested below employ elasticity to simulate very realistic calving of ice. So we ite those as examples that show the potential for realistic calving simulation when elasticity is accounted for; obviously the spatio-temporal scales we are interested in are those which can resolve accumulated effects of brittle failure: or calving fronts on the yearly to decadal time span.

But we tempered the statement anyway.

**R1:** page 4, line 14: some words in the introduction about particle models or discrete element models would be interesting (e.g. Bassis and Jacobs, 2013; Åström and others, 2014), and how they compare to the present approach.

AR: Ok.

**R1:** page 4, line 17: the main issue of using a Lagrangian approach in glaciology relies in accounting for the in/outcoming flux of ice on the domain boundaries (accumulation and/or ablation on the surface, melting/accretion at the base). You should mention in the manuscript how this problem is (or will) be overcome for realistic applications.

AR: Ok.

**R1:** page 4, line 19: FS models neglect acceleration because it is completely negligible for the time step of interest of many applications. In the proposed applications, it would be interesting to document the relative contribution of acceleration in the total momentum. Their importance, as stated here, has still to be proven?

**AR:** Agreed: their importance, as stated here, has yet to be explicitly shown. However, the particle models suggested in this review nicely capture the dynamics of calving: these models account *entirely* for acceleration. But we believe that the proportional importance of dynamic and static formulations of momentum conservation for calving applications are best left to future work, as this paper's scope is limited to rheological choices. Further, while not applied to ice directly, Choi et al. [2013] discuss the range of quasi-static damping parameters employed in DES in much greater detail.

**R1:** *page 5, line 5: avoid repeating "of ice".*

AR: Ok.

**R1:** page 5, line 7: from my experience, a Dirichlet BC is only required where you have an output flow and not on all the boundaries, as it seems the case here. Does it come from the Lagrangian formulation?

AR: The mobile nature of the mesh – that all nodes are free to move and can be deleted or added– is why we prescribe Dirichlet conditions. This model was developed to simulate very large strain problems in elasticity, and this is the mesh required for such a problem.

**R1:** page 5, line 13: This sentence is not clear and looks technical more than related to the physics in the model? Which equation is solved for incompressibility should be given here, whereas how it is solved should be given in the following section.

- AR: Ok.
- **R1:** page 5, line 23: it is not clear if the floatation is fulfilled for the floating part?

**AR:** We clarified the language.

**R1:** page 5, line 25: you mean an explicit time-stepping scheme?

**AR:** Clarified: explicit (in time), finite-element in space. That is, explicit time integration, finite element method.

**R1:** page 6, line 6: are given in Choi et al. [2013].

**AR:** "mass scaling technique that is detailed in Choi et al., 2013"

**R1:** *page* 6, *line* 14: *no need to define again the minimum element facet length.*

AR: Ok.

**R1:** page 7, line 25: I don't really think this list of capabilities is relevant for the present paper

AR: We strongly believe they are: that these experiments presented in Choi et al. [2013] is proof to the reader that the model numerics have been verified and validated. The reader of a numerical modeling paper should care that a model has passed its required benchmark tests. But to un-clutter the manuscript we have moved this albeit simple statement to Appendix A.

**R1:** page 8, line 4: we divide this section (delete the)

AR: Ok.

**R1:** page 9, line 4: the two order of magnitude differences in term of stress certainly is the result of the very particular boundary conditions applied here and therefore no real conclusion can be drawn from this setup regarding a realistic case (see major remarks).

- AR: See new Figure 5.
- **R1:** page 9, line 5: (Figs. 3a and b)
- AR: Ok.

AR: Ok.

**R1:** page 9, line 16: So why not showing these better results obtained with an higher resolution? In any case, a sensitivity study of our results to the mesh resolution would clearly improve the strengh of the paper.

**AR:** Agreed; we now present results for halved resolution.

**R1:** page 9, line 25: How much the spacing shown in Fig. 4 is dependent of the mesh. In other words, do you get the same spacing with a mesh with halved elements?

**AR:** Also see major comment response.

**R1:** *page* 10, *line* 23: *The most appropriate variable to write a criteria for damage would be the Cauchy stress, not the strain or strain-rate.*

**AR:** Duddu et al., 2013 (GRL, reviewer 2 is a co-author) use a critical strain (p. 964).

**R1:** page 12, line 9: How would you account for basal melting in a Lagrangian model?

**AR:** Basal melting is often reported in the literature in terms of meters per year of loss. We admit our implementation of this is unsophisticated and remains to be developed further. As yet, we (would) apply a Dirichlet velocity condition, in meters per year, which moves nodes vertically at those melting rates – effectively thinning the tongue. This does not change the shape or sharp-ness of various features, as might be expected in nature, or as is examined in

**R1:** page 9, line 9: (Figs. 3e and f)

more detail in numerous papers. In any case, the effect of melting is simply not the focus of this paper.

**R1:** The basal geometry from Fig. 3 does not correspond to the setup presented in Fig. 2a. In Fig 2a it is a straight line over 10 km whereas in Fig 3 there is two lines that define the base (over the same 10 km)? I am wondering if the 10 km scale indicated in Fig. 3 is therefore correct? An horizontal scale in Fig. 4 would be helpful for the same reason.

AR: The geometric setup in 2a indicates that ice is advected down a 3 degree plane until it reaches 10 km in the domain, at which point it is forced flat. Not sure where in Fig. 3 you are seeing two lines other than the 3 degree plane leading to a flat plane after 10 km. The length of the ice is also 10 km long. The 10 km scale is correct.

We added a horizontal scale bar to Fig. 4.

**R1:** The geometry in Figs. 5 and 6 look very mesh dependent and it would require some convincing arguments (i.e., a mesh sensitivity study) before moving to physical explanations about these modeled features as done in Fig. 7.

**AR:** See response to major comment.

References

**R1:** Åström, J. A., D. Vallot, M. Schäfer, E. Z. Welty, S. O'Neel, T. Bartholomaus, Y. Liu, T. Riikilä, T. Zwinger, J. Timonen and others. 2014, Termini of calving glaciers as self-organized critical systems. Nature Geoscience, 7(12), 874–878.

AR: Ok.

**R1:** Bassis, J. and S. Jacobs. 2013, Diverse calving patterns linked to glacier geometry. Nature Geoscience, 6(10), 833–836.

**AR:** It's already there.

[We have bolded questions to help ease the reading here, and broken apart the referee comment to make clear our responses to individual questions within the discussion.]

**Author response to reviewer 2, Jeremy Bassis:**

**R2: 1.** *Rheology and yield relations.* I would like to see a much more detailed description of the spectrum of rheologies and yield relations used. The authors provide a description of the usual power-law viscous creep deformation glaciologists are used to, but few equations describing the rheology beyond this. I recognize that the model used is fully documented in prior publications. However, the authors are introducing concepts that are new (or at least less familiar) to glaciologists and some hand holding is appropriate. There are also some details that are missing. For example, the 2D viscoelastic simulations are presumably done under plane stress or plane strain conditions, but I could not find which in the manuscript. (I apologize to the authors if I missed this in the manuscript.) **More importantly, I would like to see equations describing the yield relations and some description of the assumptions.**

**AR:** We include an exhaustive exposition of all rheological assumptions and flow laws now in a new section 2.2.

For example, the authors state that they use a Mohr-Coulomb yield strength. The typical interpretation of the MohrCoulomb yield strength is that materials fail when the maximum shear stress exceeds a threshold that depends on the normal stress and a cohesion parameter. This is occasionally interpreted as the initiation of new faults or the re-activiation of previously existing faults. Which interpretation are the authors assuming? Or does it not matter?

AR: It can be both. At the beginning of the simulation no plastic strain has accumulated, so prior to any failure (the ice is truly virgin) exceeding the failure threshold represents the initiation of new 'faults,' however throughout the model run previously broken areas can accumulate more plastic strain provided the failure thresholds (now detailed exhaustively) are met. We make explicit this interpretation in section 2.2 now.

**Also, what happens above the yield strength?**

**AR:** Material follows plastic flow law (new section 2.2).

Does the yield strength denote a boundary between flow laws, as in a Bingham plastic? What happens once ice has failed? Does it return to behaving like intact ice if the stress decreases beneath the yield strength (as is true in a granular material) or does it continue to behave as dam aged ice once yielded, irrespective of the current state of stress?

**AR:** Once ice is broken it is broken: no healing occurs in this rheology. So an element that has reached brittle failure continues to be evaluated as elastic (and can break further if the tresses reach MC threshold) but it is no longer evaluated as Maxwell. We arrived at this by way of the calibration experiments that indicated that we could only reproduce the strain-time curves with this requirement.

Another question I have relates to tensile versus shear failure. For example, typically, we think of crevasses as tensile failure features, but the Mohr Coulomb failure envelope is usually applied to shear failure. (In the absence of a cohesive strength, a MohrCoulomb failure law implies no tensile strength.) **How do the authors simulate tensile failure as opposed to shear failure? Are there different yield strengths used?**

**AR:** We have a Mohr-Coulomb envelope with cohesive strength and are often in the tensile region in the shallower ice depths. So we accommodate both tensile and shear failure, and this is now explicitly apparent in Section 2.2.

Typically, faulting is more important in the Earth, but in **ice people often focus on tensile failure. (We partially dispute this. (us too!)** See for example, Bassis and Walker, 2012, Proceedings of the Royal Society.). Moreover, failure envelopes in compression and tension are usually very different with compressive strengths much larger than tensile strengths. Is this accommodated in the model? Is compressive failure considered negligible? AR: We do not model compressive failure, and believe that at least for our simulations (where there are no pinning points – for instance) compressive failure is negligible.
This is stated explicitly now (Section 2.2, and in Appendix A).

There are also some technical questions associated with simulating yielded ice. We (and others) have found that the maximum shear stress criterion associated with Coulomb-like failure can be difficult to implement numerically. Instead, we (and many others) often prefer to use the effective stress (2nd deviatoric stress invariant). This is qualitatively similar, but corresponds to a Drucker-Prager granular material and not a Coulomb-granular material. I assume the authors are using the Coulomb criterion, **but do the authors need to stabilize the method to avoid the numerical errors associated with the nonrobustness of finding a maximum?**

**AR:** We use an explicit (shown in great detail now) formulation that guaranties that the failure occurs at the max.

All of these questions leave me with an imperfect understanding of the physics assumed by the authors and this clouds my understanding of the results that follow from these assumptions. I suspect most readers will have similar questions and it will help tremendously if the authors step us through the assumptions and assumed physics instead of rushing us through to the results. In many ways, I think the physical model has much greater value than the preliminary results so I urge the authors to take the time to explain the model thoroughly to the audience.

**AR:** Valid concerns and astute questions all.

- 1. See new Section 2.2
- 2. See new Appendix A

**R2: 2.** *Boundary conditions.* The authors specify velocity boundary conditions at the left, bottom and right edges of the domain. Specifying a velocity boundary condition at the bottom is a bit odd. Typically, we would specify a sliding law or, alternatively no-slip or free-slip boundary conditions. I'm a little bit worried that the velocity boundary condition contaminates the results. I would recommend either re-running simulations using a sliding law. We often like to do both the free-slip and no-slip conditions to bracket behavior when doing idealized experiments where we don't want to specify parameters in a sliding law. If this is unfeasible, then I think some additional justification for the boundary conditions is appropriate. If the authors maintain the velocity boundary condition the authors should plot basal shear stresses? The authors also might want to consider using a free-slip boundary condition for the vertical displacement in the left side of the domain. This will avoid the weird abrupt decrease in ice thickness near the left wall.

**AR: Good point.**

We show now more of our motivation for Experiment 1, and opted for your suggestion to run ome cases (now either shown or touched on in manuscript):

1. Experiment 2: purely ductile and brittle ice, no-slip / free-slip: these show that the velocity BCs do not contaminate the stress field, and that we can believe the stresses we see in Experiment 1

2. Experiment 2: semi-brittle ice, no-slip

**R2: 3.** *Model numerics and comparison with existing solutions.* The model that the authors are using is a complex viscoelastic model used to study solid Earth deformation. The model appears to have been well benchmarked agains standard solutions and so hopefully the model numerics is well understood. However, there are aspects of the numerics associated with the flow of ice that are not as well represented in the previous set of benchmark experiments. In particular, the mass weighting and damping to obtain stable solutions in the explicit integration of the Navier-Stokes equations (with inertia) does not appear to have been calibrated with ice is mind.

AR: This damping scheme does not depend on material properties specific to tectonics; rather, it is a numerical technique employed based on the characteristic speed of the phenomenon that the user wishes to resolve. But we include now in Appendix A the results of a validating experiment wherein we tuned model parameters to reproduce strain- and strain-rate-vs-time curves for laboratory prepared ice, in essentially the same exercise as in Duddu and Waisman, 2012. Reproducing this behavior in our semi-brittle ice required a great amount of parameter suite exploration, including the mass weighting and characteristic speeds, as well as exploration in the ductile to brittle strain rate threshold. Truly, the strain-time behavior of this semi-brittle rheology is sensitive to parameters, and our matching the strain- / strain-rate-vs-time behavior should give the reader some assurance that the parameters used in the idealized experiments are those which give the most realistic representation of ice behavior that we are able to reproduce.

This raises some questions about the appropriateness of the numerical parameters. The mass weighting method that the authors use to time step the Navier-Stokes equations is one of the those methods that gets periodically rediscovered. I would **personally prefer if the authors made it clear that the mass** weighted explicit integration is used as a means of avoiding the cumbersome and expensive task of solving of large-non-linear sets of equations and that individual time steps do not provide accurate solution to the equations of motions. The hope is that over long time scales the solution is approximately steady-state, which corresponds to the Stokes equations that the authors rely want to solve. Presumably, one could use, say, a multi grid or other fancy numerical solver instead to find the solution to the elliptical set of equations. Having said this, **it would be nice if the authors could show that the model that they use is able to reproduce existing analytic or benchmark solutions for glacier flow.**

**AR:** So noted. As an aside, we disagree that this method does not "provide [an] accurate solution to the equations of motions": e.g., Hughes [2000], Detournay and Dzik [2006], De Micheli and

Mocellin [2009], Choi et al. [2013], Ta et al. [2015], Lavier and Wu, [2016], to list a scant few (cited within), show that these techniques do provide accurate solutions to the equation of motion.

There have been a number of model inter comparisons that the authors could consider. I'm agnostic to the choice, but it would be reassuring to show that under viscous conditions, the authors can reproduce standard solutions for velocities and ice thickness. The authors have probably already done this and so a few sentences or a section in an Appendix would be all that is required. If possible, it would be great to see some convergence studies to show that the results shown in the paper are not numerical artifacts or signs of instabilities. The figures in the paper show jagged ice shelves. I suspect that failure will look more realistic if the authors conduct higher resolution model runs.

AR: These are all really important aspects of model development, verification/validation, and presentation. You are correct in assuming that we have explored the ISMIP-HOM suite of experiments. Unfortunately, because DES' mesh is completely mobile, it is impossible to apply the periodic boundary conditions necessary to validate the model against Experiment F in Pattyn et al. [2008; ISMIP-HOM]. However we are able to reproduce Experiment E (Haut Glacier d'Arolla) with some success. This is shown in Appendix B.

1. See new Appendix A: material and numerical parameters that reproduce laboratoryderived strain-time curves reported by Mahrenholtz and Wu, 1998.

2. See new Appendix B: Arolla Glacier benchmark experiment reproduced (from Pattyn et al., 008, I SMIP-HOM).

**R2: A.** *Interpretation of model results:* One of the most intriguing results that the authors obtain is that they produce basal crevasses under ice shelves. We tried to explain these features in a recent paper using a perturbation approach (Bassis and Yue, 2015, EPSL). We focused on viscous instead of brittle ice and found a long wavelength instability that could result in wide basal crevasses so long as the stress was sufficiently large compared to the confining pressure. In our formalism, we can also examine brittle failure by taking the limit that the flow law exponent (n) tends to infinity. When we do this we find that the dominant wavelength is of the order of the ice thickness. The growth rate of perturbations, however, becomes extremely large. This is a consequence of the fact that in our model, we assume the ice is isothermal. This implies that over long wavelengths, the strain rate and deviator stress are both constant with depth and the entire ice shelf reaches the yield strength at the same time. **This raises the question of whether the results here are consistent or inconsistent with our (admitted) limited) analytic result? If not, what controls the rate at which brittle failure propagates. What control the spacing between basal crevasses? Incidentally, the perturbation analysis that we conduct is analogous to some of the original perturbation calculations to explain boudinage in rock by Smith and others.**

**AR:** These are important questions; we're glad you brought to our attention your formalism, and in the manuscript now we engage with a comparison – albeit briefly. We thought that it was

important to show these results to the community so that questions such as the rate of brittle failure, the spacing of boudins may be addressed by the community and in our future work. We explicitly admit to the limitation of our work and that many remaining aspects need to be addressed in future work.

R2: 5. Clarification of the role of elastic stresses: The authors make a really interesting point that despite the fact that elastic stress decay over long time scales, the fractures that result from elastic stresses remain important. Based on this, the authors argue that we need viscoelastic rheologies to accomodate failure. I don't disagree with the authors. However, if elastic stresses are important (through their role in promoting failure) then, unlike purely viscous flow, simulations become an initial value problem. What I mean by this is that in purely viscous flow we can initialize a model with an unrealistic initial condition. The unrealistic initial condition will generates shocks in the model that will relax over time and we typically either initialize a model in such a way as to not generate shocks or allow the model to spin up until those shocks have sufficiently dissipated that the model is no longer contaminated by these shocks. In a viscoelastic model with failure, it seems possible that the template for failure will be strongly controlled by the initial condition -- especially if the initial condition is unrealistic and generates shocks. The authors are starting with simple wedges and allowing them to evolve. Do the authors obtain similar results if the model is first spun up to a quasi-steady state consistent with purely viscous flow and only then is failure allowed to occur? Do elastic stresses remain important if the model is started from a configuration in which elastic stresses have already decayed? What is an appropriate starting condition for models or is the initial condition not that important?

**AR:** This is a very nice point, and while not the focus of this paper, we note that we did in fact run experiments (not shown in the manuscript because they look exactly the same) where we initialized semi-brittle rheology but held the geometry in place to allow for elastic shocks to dissipate. These resulted in exactly the same failure pattern and geometry of the floating tongue shown in the manuscript. To look at the rate of fracture propagation in the context of elastic shocks and competing viscoelastic damping we need to implement an adaptive time stepping scheme that depends on occurrence of brittle failure and initiate the model with preloading. This is most assuredly the scope of future work.

**Incidental comments:**

**R2:** Page 3, near line 5 "Ductile fracture is initiated by the formation of distributed voids that eventually coalesce to form a macroscopic fracture". Laboratory experiments indicate that ductile failure growth through the nucleation and growth of voids does not occur in ice. Fractures instead usually propagate through the formation and propagation of micro-cracks. I think that is what we proposed occurs ahead of the rift in the Amery Ice Shelf. The void growth mode of failure occurs in metals (perhaps rocks as well?), but to my knowledge is inapplicable to ice under terrestrial conditions. There is, of course, the separate question of whether the macroscopic behavior of ice in glaciers can be simulated using a

framework appropriate for ductile failure of metals. However, I would like the distinction to be made more clearly in the manuscript.

**AR:** This is an incisive point (we were attempting to say as much); regardless, clarified.

**R2:** Page 8, left, right and bottom velocities are set to 300 m/a. First, I recommend using more physical notation, like inflow, outflow and basal boundary conditions, including left, right, bottom as the authors see fit. Second, the fact that the velocity is constant implies no bulk extensional stresses, which seems odd for a glacier. I would appreciate more description for the motivation for this set of experiments. I'm less confident for the evidence of a sharp brittle-ductile transition at a critical strain rate. We clearly see tensile fractures at a range of strain rates, with the controlling variable usually stress. Of course, stress and strain rate are interchangeable if the ice is isothermal, but that is not often the case.

AR: We agree: replaced left/bottom/right with inflow/basal/outflow. And yes, we also agree that ot allowing any bulk extensional stress (or strain, we prefer to think) is odd for a glacier. We clarify that this is not a glacier: we're only setting up this scenario to see how these two rheologies undergo a bending moment, admitting that almost no extrapolations that can be made from this experiment to features of interest (like basal crevasses) in an actual glacier. Experiment 1 motivates Experiment 2: Experiment 1 shows us vertical, localized, regular failure, which motivates semi-brittle ice as a rheology (since we want to reproduce vertical, localized, regular failure in a \*\*only\*\* slightly more realistic setup – Experiment 2).

**R2:** Page 1 Line 15: "We find that the use of a semi-brittle constitutive law is a necessary material condition to form the . . ." I believe necessary should be replaced with sufficient. I don't think the authors have proven that no other conditions are able to reproduce fields of basal crevasses. What they have demonstrated is that a brittle rheology is sufficient to produce this feature.

**AR:** Agreed and changed.

**R2:** Page 3 Line 5: Usually brittle failure of ice is thought to be a consequence of high stresses rather than strain rates. See, e.g., Vaughan, Journal of Glaciology, 1993 "Relating the occurrence of crevasses to strain rates".

AR: Rephrased.

**R2:** Page 3 Line 5: The point about ductile failure versus brittle failure is subtle. The coalescence of voids to form macroscopic fractures might actually be brittle. At the very least, the formation of these voids appears to be seismic. But the brittle failure that occurs may act like plastic or ductile failure over macroscopic length scales.

**AR:** Trenchant comment; we agree that macroscopic failure may be approximated by plastic or ductile failure, and say that explicitly instead.

**R2:** Page 3, Line 20 It seems odd to claim that models based on Linear Elastic Fracture Mechanics do not predict the correct stresses if their rheology is assumed to be purely viscous. By definition the "E" in LEFM corresponds to elastic so how can the rheology be assumed to be purely viscous?

**AR:** You're right: it's odd. Reviewer 1 had qualms with this statement and we rephrased accordingly.

---

## Author Response (AR2)

"Semi-brittle rheology and ice dynamics in DynEarthSol3D"
L. C. Logan, L. L. Lavier, E. Choi, E. Tan, G. A. Catania

The authors truly want to express their gratitude for the time that the reviewers took in re-reading this manuscript. We again have largely engaged with their desires to see further clarification in remaining spots of the paper, and believe that these additions have strengthened the manuscript.
Thanks again, and best regards,
the Authors
* * *
*Referee comment*
Author response

*Reviewer 1*
*I thank the authors for this new version which has clearly been improved since the first one. Nevertheless, I still have one major concern with the results presented in this new version and also many suggestions that could still improve the paper. A large part of the corrections listed below could have been avoided by a careful review of all authors of the paper.*

*My major concern is about the such large difference in effective stress given by the brittle and ductile solutions, and it would be nice if the paper could discuss more deeply this result. If the inertial term in the momentum equations were neglected, then the static equilibrium implies that the integral of the normal Cauchy stress on the domain boundary equals the integrated gravity acceleration over the domain. With such large differences in the values of the computed effective stress (more than one order of magnitude), I don't see how the integral of the Cauchy stress could be similar for the two methods. I wrote "similar" as I don't expect them to be equal since you are accounting for the inertia terms in the momentum equation, but in the other hand, I don't expect the inertia terms to contribute significantly to the total balance. This result, which is one of the major result of the paper, should be explored in more details, and the reason of such large difference explained by adding supplement information regarding Cauchy stresses and the relative contribution of inertia terms in the momentum equation.*
We agree that this is an important result and discuss it more now. The difference between the ductile and brittle stress results essentially from the fact that elastic stresses in viscoelasticity are constantly relaxed while the in the elastoplastic they are not. These can be estimated via equations (now) 8 and 9: with the values we use from the literature for bulk and shear moduli, and the range of strain rates encountered in ice, it can be shown that the elastoplastic stress is always greater than the viscoelastic stress for strain rates greater than $10^{-11}\,s^{-1}$.

*Other remarks (line numbering from the new submitted version) :*
*- in the introduction, some wording are still a bit naive or too much affirmative, and*

*sometime even not correct. For example, page 2, line 3: "these are often ... full-Stokes (FS) equations" i snot correct. There are much more lower-order models than Stokes models (see the class of models in the papers by Pattyn on ISMIP, MISMIP or MISMIP+ exercices, there is only one or two Stokes model that have applied). An other example: line 7, "most models are designed largely for steady state flow" is just a wrong statement (obviously I missed this in my first review). On what is based this strong statement regarding the fact that most model are diagnostic and not prognostic. At least, all models that performed ISMIP-HOM test F, MISMIP and MISMIP3d are capable to evolve the geometry. All this introductive part regarding ice flow models should be seriously corrected.*

We change the language describing the time integration design of many models. Regarding, however, the dimensionality of the stress tensor for other frequently-used models: in the very next sentence we say that "Models based on shallow ice (SIA) and shallow shelf (SSA) approximations of the FS equations are also in wide use and simulate ice flow well in most areas (e.g. Winkelmann et al., 2011; Lipscomb et al., 2013)." How does this statement ignore lower-order models?

*- page 3, line 1: A is not a relation, A is a fluidity parameter, which dependency to temperature is given by an Arrhenius relation.*
Agreed.

*- page 3, line 4: to my point, more than high strain rate, it is high stress or at least you should mention also or high stress*
Agreed.

*- page 3, line 5: of brittle or ductile deformation -> of ductile and brittle deformation. More logical to invert with the two sentences that follow and also I think it should be and (and not or) as the calving is the result of these two processes.*
Agreed.

*- page 3, line 17: these models they often -> these models often*
Done.

*- page 4, line 6: magnitude larger -> magnitude much larger*
Done.

*- page 4, line 11: brittle or ductile -> brittle and ductile*
Done.

*- page 4, line 25: finite element method -> finite element model*
Done.

*- page 5, line 4: the system is not complet and cannot be solved with only these*

*equations (4 unknown u, v, w and p for 3 equations). You should add the equation that you solve to close the system to be consistent.*

We clarified: when solving Stokes equation with incompressible fluid, the pressure and velocity are decoupled. Hence, pressure becomes an independent variable.
In our equations the material is compressible. We state that this is not the typical assumption in FS/SIA/SSA models. The pressure is derived from the elastic deformation of volume change and is not an independent variable.

*- page 5, line 14: it is a strong assumption, only valid if the base is temperate. Else it should be a heat flux. Anyway, my understanding is that for the test presented the temperature field is imposed (at least for experiment 1, page 13, line 19 and seems not specified for experiment 2)? If this is true, than this part regarding the temperature solution could be removed.*

We note that heat fluxes are typically applied, and clarified how temperatures are set in both experiments. However, as this paper presents the model's capability, we believe including a short description on its solution to the conservation of energy is pertinent. And, while we do not change the applied temperatures at the boundaries throughout time, the non-isothermal temperature field is a part of the nonlinear viscosity. We believe this impact means that the temperature component should be kept in the methods section.

*- page 5, line 20: what does it mean to enforce mass conservation via elasticity? Which equation are you solving for? See also the point page 5, line 4.*

We have clarified: DES does not solve or enforce the conservation of mass equation.

*- page 5, line 22: accounting for accumulation or ablation is not done through a Neumann conditions. Neumann conditions for the Navier-Stokes problem allow to enforce stress type conditions.*

Deleted.

*- Equation (5): $tr(\epsilon)=\Delta V/ V$ and should be 0 for an incompressible material like ice. Or do you account for the void volume introduced by the opening of cracks?*

We do not account for void volume. We simply do not assume ice is incompressible.

*- Equation (7): not obvious to derive (7) from (6), may be a reference or more explication could help the reader? Also, the notations are not consistent, with sometime VE and other ve for the viscoelastic indice on stress.*

Added extra step.

*- Equation (8): again, the trace of the strain rate is zero for an incompressible material like ice*

We do not assume incompressibility.

*- page 7, line 20: I am confused with the sign of the yield surface criteria. The elastic stress is on or within the yield surface if $f \le 0$?*
Fixed.

*- page 8, line 6: $\sigma_1$ is not used in Equation (9). Should be defined below Equation (10).*
Fixed.

*- Equation (17): Is $K_s$ different than the $K$ parameter defined above?*
Fixed.

*- page 13, line 14: boundary conditions in glaciers are often either formulated as a Weertman-style sliding law -> boundary conditions at the base of glaciers are often either formulated as a sliding law*
Done.

*- page 13, lines 17-19: this sentence is not clear. You should insist that you are solving a Lagrangian problem (which is not the classical approach in ice flow modeling, most model being using an Eulerian approach). Suggestion: During the transient simulation, when a basal node reaches the position x=10km, its velocity is tilted to an angle of 3 degrees*
Clarified.

*- page 15, line 12: the brittle ice remains largely or exactly in its initial configuration? The largely is to said that the deformations are only due to elastic deformation and are small regarding to plastic ones after one year? May be you should be more precise here.*
 Clarified: added precision.

*- page 15, line 14: it should ne mentioned that these results are not shown?*
It is shown: Figure 5.

*- page 15, line 17: the definition of semi-brittle has already be given above. See also the next point.*
Deleted.

*- page 15, lines 19-26: this text should be in the model presentation*
Removed this lengthier discussion of damage mechanics as it is not pertinent to the presentation of DynEarthSol.

*- page 15, line 25: I don't think that strain can be used to estimate when ice should start to accumulate damage. In ice sheet for example, you can find ice that have undergone very large strain that are not damaged at all. On the contrary, the fact that damage should be initiated using strain-rate is still debated in the community, as stress might be*

*a much pertinent variable for that. A lot of the material presented on this paper rely on the work by Duddu et al., 2013, but it was controversial as stated in the comment by Gagliardini et al. (2013).*

We have taken out the section that contains this. By the way: thank you for drawing our attention to that *Correspondence* in *J.Glac.*; it is an illuminating commentary of which we were previously unaware.

*- page 16, line 10: I am wondering how the stresses (deviatoric and Cauchy) resulting in this procedure looks like. Are they continuous between ductile and brittle area? How large are the discontinuity? How the choice between ductile and brittle distributes over the domain. May some more results on this could be added and useful results on how the model behave might be obtained from this analysis?*

They are not continuous (if by continuous you mean that some portion of a single element is ductile while the other portion is brittle). What distinguishes ductile from brittle -- how this looks throughout the domain -- is seen readily from the strain rate plots. Strain rates $< 10^{-7}$ s$^{-1}$ are ductile, and otherwise brittle. The stresses resulting from this threshold can be seen by comparing the effective stress field against the strain rate field. These are both shown in the figure for the semi-brittle experiment.

*- page 16, line 25: 2 difference mesh sizes -> 2 different mesh sizes*

Fixed.

*- page 18, line 11: we determined an accumulated -> we determined that an accumulated*

Fixed.

*- page 18, line 12 (and at many other places in the manuscript): .03 should write 0.03*

Fixed.

*- page 19, line 6: I would said that one hour and even 24 hours is a small computational cost, so the argument of too large computational cost for finer resolution is a bit odd?*

The quartered resolution requested in the previous review of this paper would have taken about 8 days. This we found too costly given the limits on response time.

*- page 19, line 12: is consistent with and predicted by rate-independent. Missing a word at the end of the sentence?*

Fixed.

*- page 19, line 13: one developed should be suppressed*

Fixed.

*- page 20, line 13: for a Stokes model, the evolution of the GL is not necesserally based on a criteria on ice thickness but can solve a proper contact problem (so e.g. instead of*

*i.e. would be more appropriate here)*
Fixed.

*- page 21, line 6: 3 difference stresses -> 3 different stresses; and Figure 10a and b -> Figures 10a and b*
Fixed.

*- page 21, line 10 (and elsewhere): Fig. 10e and f -> Figs. 10e and f*
Fixed.

*- Figure 3: two many text in the caption. These comments should be (and are already for some of them) in the main text. The pink vertical line of Fig. 4 should also be added on Figs. 2 and 3 to show where is the GL precisely.*
Fixed.

*Reviewer 2:*
*The semi-brittle rheology employed by the authors of this study is novel and has the potential for significant improvement of our understanding of glacier failure. I think the*

*authors have done an excellent job responding to reviewer comments and I generally prefer to avoid torturing authors with a second round of changes. I still, however, I have a small number of minor comments, described below, that the authors may consider addressing.*

*Comment 1: I am satisfied by the updated description of the rheology. I would prefer to see the rheology cleanly separately from the numerics because the rheology defines the "physics" and is at the heart of the model while the numerics consists of a set of choices made to solve the physical equations, but I leave this at the description of the authors. The one question I have is whether the Lagrangian scheme is valid for large strains, which is important for the deformation of ice over long periods of time. This should be clarified.*

The Lagrangian and Eulerian schemes are concerned about whether to describe motions in terms of reference or spatial coordinates, which is independent of whether to allow large strain. We clarify that DES can *accumulate* large strain by adding up small strain increments, and the use of small strain at each time step is justified by a relatively small time step size.

*Comment 2: I still object to the term "dynamic relaxation" being described as a means of accurately solving the dynamic (i.e., with inertial terms) equations. In the original experiments by Cundall and others, dynamic relaxation was clearly introduced as an approximation. In fact, it is straight forward to prove that the mass weighting approximation can be arbitrarily incorrect when applied to the full dynamic problem and only approximates the inertia-less solution in the limit of long time scales. To sketch out the proof for the first case, we note that a solution to the short-time scale elastic behavior consists of an elastic wave traveling at the p-wave velocity, which we call c. In the mass weighted method, the p-wave velocity is c', where c' is less than c. After some time t, using d'Alembert's solution, an arbitrary perturbation will have travelled a distance d=c*t. In the mass weighted system, the perturbation will have only travelled a distance d'=c'*t< c*t. Given the fact that the perturbation is arbitrary, the mass weighted solution can be arbitrarily incorrect unless c' approaches c. This completes the proof that dynamic relaxation cannot provide an accurate solution of the full dynamic equations. We next turn to the limitations of applying dynamic relaxation to the quasi-static analogue (i.e., when inertia is omitted). The quasi-static limit corresponds to solving an elliptic equation in which velocity is non-local. In the dynamic, mass weighted approximation, one still solves a hyperbolic or diffusive set of equations in which information propagates locally. Let us again consider a perturbation, but this time to the ice thickness. In the quasi-static limit, a point a distance "x" away from the location of the perturbation feels the effect of the perturbation instantaneously. However, in the dynamic limit with mass weighting, information still only propagates at a speed of c'. Hence, for sufficiently small time "t", c'*t<*

We accept this demonstration and accede to stating that the dynamic relaxation method only approximately solves the dynamic equations.

*Minor comments:*

*Page 20: Matching the limited laboratory data to simulate ductile failure is a process fraught with error. First, the laboratory data are generally obtained using laboratory grown samples of ice and these samples do not necessarily reflect "actual" ice. Second, the studies seem to lump many different crystal deformation processes into a "damage" parameter and this is not always robust. However, given the limited laboratory and field data available, this is probably the best that can be done. This limit afflicts previous studies, like those of Duddu and others that were also limited by the lousy quality of laboratory studies.*

We agree and note the limitation now.

*Page 31: the lack of numerical convergence in Experiment 2 is worrisome. Is the spacing of basal crevasses a function of grid spacing does the model simply require a much greater resolution than the authors are capable of providing to numerically converge? This is something that the authors should comment on.*

We agree that -- while we are not surprised that ice failure scales with element size -- it may be unsettling to report that boudin spacing is sensitive to resolution and unsatisfying to the reader. We stress that these geometric features are a qualitative (not quantitative) result; we find it more important that the simple setup of advecting ice flow over a jump in boundary condition promotes ice failure, and the subsequent stretching produces geometries seen in nature. We fully acknowledge the incompleteness of the exploration, as the nature of these features will be explored more fully in future work. But we stress to the reader that these are qualitative results, a consequence of rheology and boundary conditions, but certainly not exhaustive or intended to be compared to glaciers in a quantitative way.

*References cited are not all in the reference list. This should be carefully checked.*
Fixed.

*Page 16, line 9: sigma_e is the effective stress, not the effective pressure. Effective pressure is usually defined as the pressure minus pore pressure. The effective stress is the 2nd stress invariant*
Fixed.

*Page 16, line 23: Grammar problems—>"When simulating ice rupture however these models they often employ failure criteria developed with elastic underpinnings."*
*-missing commas before and after however*
*-extra "they"*
Fixed.

*Additional minor comments by editor:*
*p. 2, Line 8: I agree with reviewer 1 that probably 'most' models have prognostic capabilities by now (although they may not all have been used for this)*

Agreed and fixed.

*Fig 11 (appendix): clarify the legend in the figure, I assume 'FS' in the legend stands for 'Full Stokes'.*
Fixed.

*P. 26 line 16: it should be Murray T. (surname is Murray)*
Fixed.

[revised manuscript text omitted]
\left(\boldsymbol{\sigma}_{ep}^{t+\Delta t}\right) = f\left(\boldsymbol{\sigma}^t + \Delta\boldsymbol{\sigma}_{ep}\right) = 0 \tag{17}.$$

In the principal component representation, $\sigma_A = E_{AB}\epsilon_B$ where $\sigma_A$ and $\epsilon_A$ are the principal stress and strain, respectively, and $E$ is a corresponding elastic moduli matrix with the following components:

$$E_{AB} = \left(K - \frac{2}{3}G\right) \text{ if } A \neq B \text{ and} \tag{18.a}$$

$$E_{AB} = \left(K + \frac{4}{3}G\right) \text{ otherwise} \tag{18.b}.$$

By applying the consistency condition and using $\boldsymbol{\sigma}_{et}^{t+\Delta t} = \boldsymbol{\sigma}^t + E \cdot \Delta\boldsymbol{\epsilon}$, we obtain the following formula for $\beta$,

$$\beta = \frac{\sigma_{el,3}^{t+\Delta t} - \sigma_t}{\frac{\delta g_t}{\delta\sigma_3}} \text{ for tensile failure and} \tag{19.a}$$

$$\beta = \frac{\sigma_{el,l}^{t+\Delta t} - N_\phi \sigma_{el,3}^{t+\Delta t} + 2C\sqrt{N_\phi}}{\sum_B (E_{1B}\frac{\delta g_s}{\delta \sigma_B} - N_\phi E_{3B}\frac{\delta g_t}{\delta \sigma_B})} \quad \text{for shear failure} \tag{19.b}.$$

Likewise, $\delta g / \delta \sigma$ takes different forms according to the failure mode:

$$\delta g / \delta \sigma_1 = 0 \tag{20.a},$$

$$\delta g / \delta \sigma_2 = 0 \tag{20.b},$$

$$\delta g / \delta \sigma_3 = 1 \text{ for tensile failure and} \tag{20.c}$$

$$\delta g / \delta \sigma_1 = 1 \tag{21.a}$$

$$\delta g / \delta \sigma_2 = 0 \tag{21.b}$$

$$\delta g / \delta \sigma_3 = \frac{1+\sin\psi}{1-\sin\psi} \text{ for shear failure.} \tag{21.c}$$

Once $\Delta \varepsilon_{pl}$ is computed, $\sigma_{ep}$ is updated as

[revised manuscript text omitted]